## Registered report

psychology

executive control, training, valuation, food

**Author for correspondence:**
Lucas Spierer
e-mail: lucas.spierer@unifr.ch

# Modifying food items valuation and weight with gamified executive control training

Hugo Najberg, Maurizio Rigamonti, Michael Mouthon and Lucas Spierer

Neurology Unit, Medicine Section, Faculty of Science and Medicine, University of Fribourg, PER 09, Chemin du Musée 5, 1700 Fribourg, Switzerland

(iD) LS, 0000-0003-3558-4408

Recent lines of research suggest that repeated executive control of motor responses to food items modifies their perceived value and in turn their consumption. Cognitive interventions involving the practice of motor control and attentional tasks have thus been advanced as potential approach to improve eating habits. Yet, their efficacy remains debated, notably due to a lack of proper control for the effects of expectations. We examined whether a one-month intervention combining the practice of Go/NoGo and Cue approach training modified the perceived palatability of food items (i.e. decrease in unhealthy and increase in healthy food items' palatability ratings), and in turn participants' weights. We assessed our hypotheses with a parallel, double-blind, randomized controlled trial. Motivation and adherence to the intervention were maximized by a professional-level gamification of the training tasks. The control intervention differed from the experimental intervention only in the biasing of the stimulus–response mapping rules, enabling to balance expectations between the two groups and thus to conclude on the causal influence of motoric control on items valuation. We found a larger decrease of the unhealthy items' palatability ratings in the experimental (20.6%) than control group (13.1%). However, we did not find any increase of the healthy items' ratings or weight loss. Overall, the present registered report confirms that the repeated inhibition of motor responses to food cues, together with the development of attentional biases away from these cues, reduces their perceived value.

# 1. Introduction

Since the overconsumption of high-density energy palatable food contributes to the development and maintenance of many health disorders, including obesity [1], diabetes [2] or metabolic syndrome [3], interventions helping to reduce this behaviour are much needed. Recent reports indicate that a potential approach might consist in practising executive tasks in which motor response towards food items is repeatedly inhibited or executed. Inhibiting motor responses to energetically dense and palatable food items during Go/NoGo tasks (GNG) has indeed been shown to reduce their perceived value, consumption in bogus taste tests and participants' weight (for reviews, see [4–7]). Likewise, executing motor responses to task-relevant healthy items during Cue approach training tasks (CAT) has been shown to induce opposite effects [8–11].

The main hypothesis advanced to account for the effect of inhibitory control training on eating behaviour [7] suggests that it may reduce unhealthy food consumption by developing 'inhibition reflexes' of the motor responses to the targeted items [12,13]. In Go/NoGo tasks, participants are instructed to respond as fast as possible to a given category of items and to withhold their responses to another category. Such practice actually acts as motivational conditioning paradigm which eventually automatizes the engagement of inhibition processes via associative learning mechanisms and reduces its perceived value. When the NoGo stimuli become associated with avoidance/aversion, its presentation would directly trigger the avoidance/aversive centre, which would then suppress the activation of the approach/appetitive centre [14]. This mechanism is thought to eventually result in the decrease in the hedonic and motivational value of the NoGo stimuli [15–18].

Approach bias training is thought to modulate food item valuation by developing attentional and approach biases towards healthy food cues [19]. In CAT, items are displayed for a short period during which a Go cue prompting a motor response may be presented [8]. The task of the participant is to respond to cued items before their offsets. Since task performance is improved by paying attention to and rapidly reaching the items associated with the cues (but not to those not associated with the cue), attention and approach tendency is eventually automatically allocated to the cued items [20]. In turn, the target items' saliency and perceived value increases [21,22] as well as their consumption [23,24] (see [19] for a review).

Of note, responses to Go stimuli during the Go/NoGo task and the withholding of responses to the non-cued stimuli during the CAT may also, respectively, develop approach ([25] Experiment 2) or avoidance tendency [8–10].

On this basis, we reasoned that the development of automatic inhibition to unhealthy items and of attentional biases towards healthy items would act synergistically to eventually improve eating habits. This approach would indeed promote the replacement of unhealthy items by healthy ones and not merely reducing unhealthy item consumption, thereby ensuring to maintain satiety. This hypothesis is supported by previous findings for large decreases in reward responses to unhealthy items and body fat with multifaceted training approach involving both attentional bias modification and response inhibition approaches (e.g. [26]).

While current studies suggest that food-related interventions based on repeated motor control of responses to food items may improve eating habits, they suffer two main limitations. First, the training or task parameters in previous interventions might have limited their efficacy (see [27] for discussion): they involved short single training session [28,29] and/or used training tasks with limited individual adjustments of the target items to participants' tastes [26,30] and of difficulty levels [31,32]. While some research suggest that improving the practice environment with task gamification did not improve training feasibility, acceptability or outcome [27,32], the effect of most previous studies may still have been minimized by presenting the task in weakly motivating practice environments [27,32].

Second, current studies on food-related executive training did not fully control for participants' expectations (see [33] for discussion). Causal inferences on the effects of interventions can only be drawn when they are contrasted with control conditions differing only at the level of their active 'ingredient', which in the present case is the motor control of responses to the target food items [34]. Yet, control interventions in previous studies used non-food images [26,35,36], tasks loading only weakly on executive control [27,31,32] or tasks without inhibition towards targeted items [29,37–40]. Participants in the control and experimental group may thus have generated different expectations regarding the effect of the intervention on food-related behaviours or on executive control performance, respectively. Hence, expectations might have confounded any between-group differences in food valuation and consumption outcomes.

The present project addressed these limitations by testing with a double-blind, placebo-controlled, parallel, randomized intervention trial whether a one-month (20 min per day, 5 days per week) parallel practice of food GNG and CAT would decrease high-density energy unhealthy items valuation (as indexed by their perceived palatability), increase low-density energy healthy items valuation and in turn reduce participants' weight. The GNG training aims to reduce unhealthy items' value by associating them with motoric inhibition. The CAT aims to increase the value of healthy items by biasing attention and approach tendency towards them. We further assessed if the intervention results in weight losses and whether this physiological effect is mediated by the modification in items' palatability ratings.

Motivation and adherence were maximized by gamifying the intervention; we introduced an enjoyable auditory and visual practice environment, as well as intrinsic and social challenge and reward mechanisms. In addition, progressive difficulty levels ensured that the training tasks remain adapted to participant's performance (and improvement thereof). The interventions were also individualized to each participant's tastes and eating habits by specifically targeting their preferred high-density energy items, as measured with palatability scales at the beginning of the intervention.

To control for expectations, we contrasted the effects of the intervention with those recorded in a control group participating in the exact same intervention, except that the NoGo inhibition trials and the Go execution trials were evenly distributed between the unhealthy and healthy food items. By doing so, only the 'active ingredient' of the experimental intervention were neutralized (i.e. the biasing of associations between the unhealthy food items and the inhibition of motor responses). Hence, since in the control group, the unhealthy items were still associated with response suppression (only with a lower probability than in the experimental group) and that the mechanism of devaluation via motoric inhibition or attentional bias modification could unlikely be inferred, we consider that participants' expectations on the effects of the interventions should be similar between the two groups. To further minimize any group difference in expectations, participants in both groups were informed on the outcomes expected for the experimental intervention (i.e. that we aim at reducing the perceived value of unhealthy items and reversely for the healthy items, to eventually reduce weight). Previous evidence for an absence of effect of participants' awareness of training goals on food-ICTs effectiveness [35,38] indicate that this approach should not influence the efficacy of our experimental intervention. Furthermore, to minimize the potential effect of the larger associative uncertainty in the control than experimental group on motivation and responding [41–44], and participants' expectations on the aim of the study, neutral distractors food items were added to both the experimental and control group.

We tested a first set of hypotheses by contrasting a between-subjects factor Intervention (Experimental; Control), and the within-subjects factors Session (Pre-; Post-intervention) and Food type (Healthy; Unhealthy):

The triple interaction term assessed:

— Hypothesis 1 (H1): a larger increase in the difference between unhealthy than healthy items' palatability ratings in the experimental than control group between the pre- and post-intervention.

Two Intervention by Session interactions planned contrasts further tested:

— H1a: a larger decrease in unhealthy items' palatability ratings in the experimental than the control group between the pre- and post-intervention;
— H1b: a larger increase in healthy items' palatability ratings in the experimental than the control group between the pre- and post-intervention.

We tested with an Intervention by Session interaction:

— H2: a larger weight loss in the experimental than control group.

A mediation analysis assessed:

— H3: the shift in items' valuation (i.e. the quotient between mean unhealthy rating and mean healthy rating) mediates the change in weight.

The Stage 1 protocol associated with this Registered Report received in-principle acceptance prior to data collection and analysis on 14 November 2019. The accepted Stage 1 manuscript, not including results and discussion, can be found at: https://osf.io/2wgn7/?view_only=1bd7aacf997c4441815bbf231233db43.

# 2. Material and methods

All documents related to the intervention programme (booklet, installers, questionnaires, etc.), and all R analyses scripts and their outputs are available for download on the study OSF page (view-only link with anonymous access: https://osf.io/4a7wh/?view_only=1bd7aacf997c4441815bbf231233db43).

## 2.1. Participants

### 2.1.1. Sample size and power analysis

For Hypotheses 1, 1a, 1b and 2, our sample size was calculated using Monte Carlo simulations [45,46] (we used another approach for Hypothesis 3 since it involved a different statistical design; see scripts and figures in the study OSF page):

1. For each hypothesis separately, we determined the absolute smallest effect of interest on principled ground. We did not use previous literature to determine effect sizes of interest because the reported effect sizes are relative to each study's parameters (e.g. within-subject variance or number of trials), and may be over-inflated due to publication biases (for discussion, see [47]).
2. For each dependent variable separately, we identified the task-related between- and within-subject variance based on previous datasets from studies with the same approaches.
3. For each dependent variable separately, we generated for every sample size ranging from 20 to 60 per group (by steps of 10) 10 000 simulated datasets per condition by randomly drawing values from normal distribution derived from the parameters reported in points 1 and 2.
4. For each hypothesis separately, we extracted the percentage of $p$-value of the interaction term of interest that was below our 0.05 $\alpha$ for each sample size (i.e. the power).
5. We identified the minimal sample size providing a 0.9 power to detect our effect in the simulated data.

#### 2.1.1.1. H1: modification in palatability rating

We estimated the task within-subject variance to be of 7/100 points on the palatability VAS (conservative estimate), the between-subject variance of 15/100 (from Lawrence *et al.* [35], who used a similar task), and we consider 5/100 points as the minimal change of interest in palatability ratings after our one-month intervention for both the decrease in unhealthy and increase in healthy food ratings. On this basis, 23 participants per group (46 in total) were necessary to detect a significant interaction of the $2 \times 2 \times 2$ mixed ANOVA with a power of 0.9.

For H1a and b, the same parameters as H1 were applied on a $2 \times 2$ mixed ANOVA. In this situation, 43 participants per group (86 in total) were necessary to reach a power of 0.9.

#### 2.1.1.2. H2: weight loss

We estimated our population within-subject variance in weight to be of 1 kg, the between-subject variance to be of 10 kg (extracted from our ongoing studies with corresponding groups, DePretto *et al.* [48]), and we consider a minimally relevant loss with a one-month executive training intervention as 1 kg (maximal healthy weight loss with restrictive diets are of 0.45 kg wk$^{-1}$ [49]. On this basis, 22 participants (44 in total) were necessary to detect a significant interaction term in our mixed $2 \times 4$ ANOVA with a 0.9 power.

#### 2.1.1.3. H3: changes in palatability ratings mediate changes in weight

According to Vittinghoff *et al.*'s method [50], the minimal sample size for our mediation model to reach a 0.9 power ($\alpha = 0.05$) can be computed by estimating: the smallest $b2$ of interest (0.1), the standard deviation of the change in palatability rating (7 points within-subject variance of palatability ratings), the standard deviation of the error (1 kg within-subject variance of weight) and the correlation between the factor intervention and weight loss (0.3 average correlation size). On this basis, 24 participants in total are needed using the R powerMediation package [51].

Our final sample size thus corresponds to the highest sample size required among all hypotheses, total $n = 86$ participants. We replaced dropouts and participants excluded from the analyses to

maintain the final sample size. However, participants with missing one- and four-month follow-up weight measures were not replaced.

## 2.1.2. Recruitment and screening

Participants were recruited via public advertisement.

Inclusion criteria included:

— 18- to 45-year-old healthy individuals
— BMI > 20; as we expected weight loss, no BMI near the lower normal threshold of 18.5 was taken to avoid inducing underweight states.
— Maximally 2 points difference in attraction towards salty and sweet food on our custom palatability rating questionnaire. Since participants had to discriminate the Go and NoGo trials using a sweet/salty rule in the Go/NoGo task, we balanced the liking of both types of items.

Exclusion criteria included:

— The consumption of any prescribed medication 7 days before the first testing session and during the intervention.
— Past or current diagnosis of eating disorders.
— History of weight gain/loss of more than 10% body weight in the last six months, to minimize confounds related to a tendency for spontaneously variable weight.
— Any restrictive diet.
— No plan on actively losing weight with a restrictive diet in the next four months. Although the effects on weight reduction were observed independently of participants' intentions to diet [36], we controlled for this aspect via this exclusion criterion to minimize the possibility of being confounded by large variations in participants' motivation to control their diet.
— A previous participation to executive control training study.

## 2.2. General procedure

All procedures were approved by the Commission cantonale (VD) d'éthique de la recherche sur l'être humain (CER-VD; protocol 2019-02149). Participants came to the laboratory of the Neurology Unit, University of Fribourg, to be informed on the study and give their informed consent. We compensated participants for their participation by offering the tablet used for the training (*ca* CHF 180), transformed in cash money pro rata temporis in the case of withdrawal. Once the informed consent was given, participants filled in a custom-made general health questionnaire and a palatability questionnaire to verify inclusion and exclusion criteria. They were then assigned to one of the control or experimental interventions. This assignment was determined *a priori* on a pseudo-random list by our laboratory engineer, who did not speak to the participants. A tablet with the installed intervention condition was provided to the experimenter, who in turn gave it to the participant. The participants were then instructed on the rules of the intervention and given a tablet (model Samsung Galaxy Tab A 2019) with the intervention android application software pre-installed, and with a booklet explaining the functioning of the tasks. The participants in both groups were told that the intervention aims by playing specifically designed games at reducing the perceived value of unhealthy items and reversely for the healthy items, to eventually reduce weight. To prevent the target food item selection procedure from cuing the participants about the implicit aims of the study (i.e. modifying implicit responses to the target food items to modify their valuation and consumption in the experimental group), we informed them that we select their preferred items to increase the task difficulty and avoid ceiling effects. They were not informed about the two conditions of stimulus–response mapping rules (biased towards the target unhealthy items for the experimental group intervention versus unbiased for the control group), nor about the actual aim of the study.

Since the icon of the application is identical for the experimental and control condition, and another tablet was used to explain the game mechanics, the experimenter was not able to infer the participants' condition and both the experimenter and the participants remained naive to the condition assignment, ensuring the double-blinding.

Before the intervention, each participant reported their weight and complete analogue scales of item valuation directly on the tablet. Then, the two tasks (GNG and CAT) were unlocked, and the intervention

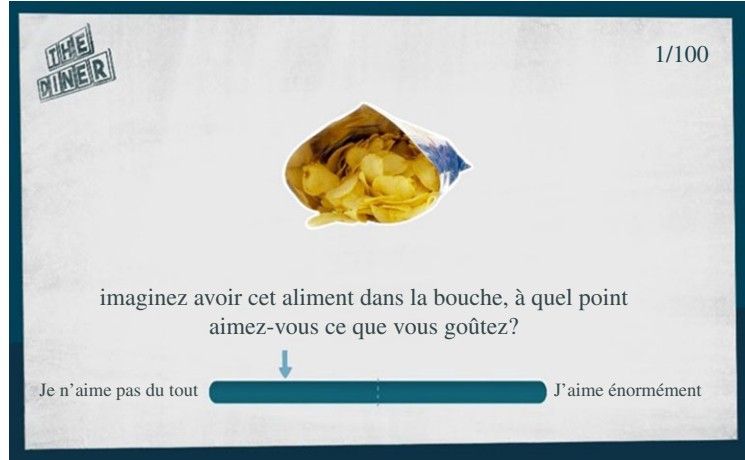

**Figure 1.** Screenshot of one item of the palatability rating questionnaire with the visual feedback of the response (the blue arrow).

was able to start. The items used during the intervention were the 50% most highly rated pictures of each category (sweet unhealthy, salty unhealthy, sweet healthy and salty healthy).

To complete the intervention, participants had to play 10 min per day at each task, resulting in 20 min of training time, 5 days a week for four weeks (20 days in total). They were instructed to distribute their daily training time as they like, but to only play when they feel rested and in a calm environment. Once the intervention was completed, participants reported their weight again and complete the same analogue scales of items palatability. They then came a second time to our laboratory to complete the core module of the Game Experience Questionnaire [52] and a custom-made debriefing questionnaire.

One and four months after the end of the intervention, we asked them again to self-report their weight.

The experimental and control groups were randomly renamed 'Cond1' and 'Cond2' by our laboratory engineer, who was not involved in the further steps of the study. The experimenter thus remained blinded up to the end of the analyses.

## 2.3. Weight

Participants were instructed to report their weight just after waking up and going to the toilets at four different time points using the same scale: the day they start the intervention, the day after completing it, one and four months after the intervention.

## 2.4. Item valuation

Analogue scales were used to assess items palatability ratings pre- and post- intervention (see Lawrence *et al.* [35] for a similar procedure). Fifty healthy and 50 unhealthy pictures (100 in total) were displayed sequentially on the tablet screen in a randomized order, with the question 'Imagine having this food in your mouth, how much do you like its taste?' (figure 1). Participants had to report their ratings on a 10 cm analogue scale with 'not at all' to 'very much' as minimal–maximal anchors (i.e. respectively, 0 and 100 points) and with a marker in the middle (i.e. at 50 points). A blue arrow indicates where the participants had responded. Post-intervention, only the trained items (i.e. those above the median at pre-intervention) were displayed again. Since the neutral items are distractors, they were not rated.

To improve the sensitivity of the ratings to the devaluation effects, participants were instructed to rate each item intuitively [53] and to complete the questionnaires just after reporting their weight (i.e. after waking up and before eating).

## 2.5. Stimuli

The stimuli are pictures selected from the Food-Pics database [54] and from freely available pictures on the Internet. The food pictures are divided into six categories based on their healthiness (healthy, unhealthy or neutral) and on their sweetness (sweet versus salty). Healthy items are defined as having a caloric density below the first quartile of the food picture database (less than or equal to

**Table 1.** Task-specific parameters.

| | GNG | CAT |
|---|---|---|
| Go/NoGo rate | 70% Go | 25% Go (cued items) |
| | 30% NoGo | 75% NoGo (non-cued items) |
| maximum stimulus duration (disappear after response) | 1250 ms | 1000–1500 ms[a] |
| feedback duration | 250 ms | |
| visual cue duration | n.a. | until item offset |
| cue delay | n.a. | go signal delay (GSD): based on difficulty level, see table 3 |
| interstimulus interval (ISI) | 1000–2000 ms | 800–1300 ms[b] |

[a]Because of typo in the Stage 1 manuscript, the stimulus duration has been changed from '1250 ms' to '1000–1500 ms'. This deviation from Stage 1 received editorial approval and was corrected before data collection.
[b]Because participants respond to only 25% of the trials during the CAT, we reduced its ISI to prevent boredom.

49.9 kcal/100 g) while unhealthy items are above the median (greater than or equal to 198 kcal/100 g). Items were considered as neutrals when they could not be qualified as either healthy or unhealthy (i.e. rice), if their eaten quantity was a major factor determining their healthiness (i.e. red meat, salt, sugar, honey, maple syrup) or if their picture included both healthy and unhealthy food (i.e. pancake with fruits pieces).

Non-ambiguous pictures were then divided into sweet or salty ones. To increase the variety items, pictures of crisps, salt, sugar, chocolate bars, soda, fruit salad and milkshake were added to the standardized database (see data in the study OSF page for details).

Based on the data from the Food-Pics database, the selected healthy and unhealthy items have mean palatability ratings of, respectively, 60.2 and 62.3 (Cohen's $d = 0.23$).

## 2.6. Training tasks

The gamified intervention is implemented as an android application developed on the 2019 version of Unity software (Unity3d.com, 2015).

Before starting a run, participants were able to freely choose between the Go/NoGo (GNG) and CAT tasks. As detailed after the tasks' descriptions, the same principle applied to each task: participants had to complete as many trials as they can in a row. Yet, each successful trial increased the task difficulty, and after a limited number of errors, the run was over.

In the experimental intervention, healthy items were always Go items and unhealthy NoGo items. In the control intervention, both healthy and unhealthy items had the same chance to be Go or NoGo items. Neutral items had a 20% chance to appear instead of a healthy or unhealthy item in both conditions.

Tasks' parameters are reported in table 1. The percentage of healthy, unhealthy and neutral items based on the trial condition and intervention for each mechanic is reported in table 2.

### 2.6.1. Go/NoGo

In the Go/NoGo task, participants were presented with food pictures and instructed to drag as fast as possible towards the bottom of the screen items of a given category. The Go category was either sweet items (e.g. orange, ice-cream; half of the training sessions) or salty items (e.g. green beans, hamburger; other half of the sessions) and the NoGo category the other. The presentation of target from each category was equiprobable. If the response is on a target item (Hit) but above the current reaction time threshold (RTT), a negative 'Too late' feedback is displayed. If they respond on a target item on time or if they withhold their response on a non-target item (Correct Rejection; CR), a positive green feedback and the awarded points are displayed. If they respond on a non-target item (False Alarm; FA) or if they withhold their response to a target item (Miss), a negative red feedback is displayed (figure 2). In total, there are 70% of Go trials and 30% of NoGo trials to influence response potency.

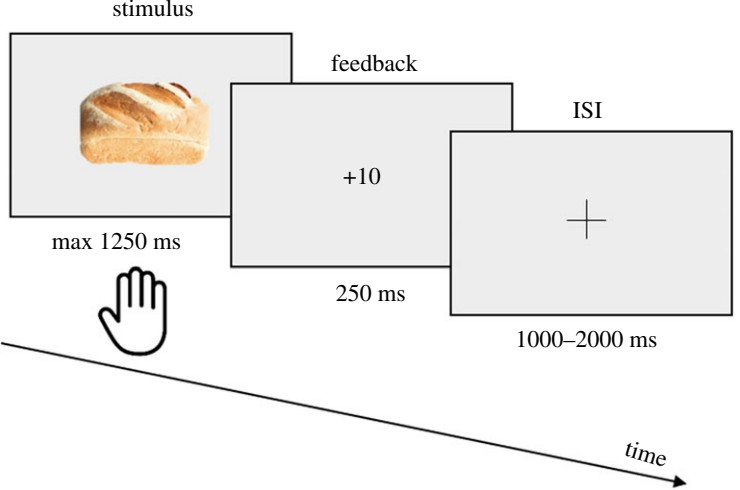

**Figure 2.** Schematic GNG task timeline.

**Table 2.** Proportion of item category displayed for each trial condition and group.

| trial condition | item type | | |
| --- | --- | --- | --- |
| | healthy (%) | unhealthy (%) | neutral (%) |
| experimental group | | | |
| Go trials | 80 | 0 | 20 |
| NoGo trials | 0 | 80 | 20 |
| control group | | | |
| Go trials | 40 | 40 | 20 |
| NoGo trials | 40 | 40 | 20 |

While feedback on performance may interact with the effect of training [55], it allows ensuring a correct understanding of the instruction by the participants and using scores as gamification parameters.

### 2.6.2. Cue approach training

In the Cue approach training task, food items sequentially appear on the screen on random predetermined locations. Participants were instructed to respond to the items only when a cue (green circle around the item and a bell sound) is displayed and before the item disappears. If they manage to do so (Hit), a positive feedback with the awarded point is displayed. If they respond on a cued item after the item disappears, a negative feedback with the mention 'Too late' is displayed. If they respond to a non-cued item (FA) or if they withhold their response to a cued item (Miss), a negative red cross feedback is displayed. If they withhold their response to a non-cued item (CR), a dark grey–green feedback is displayed with a neutral non-ascending sound and a third of the Hit point is awarded, to avoid creating attentional bias on NoGo trials (figure 3).

## 2.7. Game design

For the GNG, after six successful Go or NoGo trials, the time limit for a Go response to be considered as correct (the RTT) decreased. For the CAT, after three successful Go trials, the delay between the Go cue and the stimulus offset (the GSD) decreased (table 3). By increasing the time pressure, the task difficulty increases since this manipulation increases the probability to commit speed error ('too late' trials) and/or commission error to NoGo trials (because response prepotency increases). The 18 levels of task difficulty spanned from low time pressure (very easy) to an impossible time pressure corresponding to the minimal

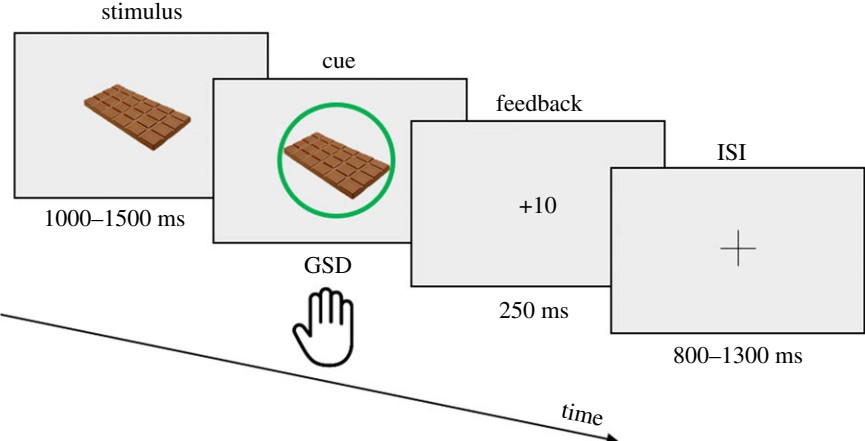

**Figure 3.** Schematic CAT task timeline. Because of typo in the Stage 1 manuscript, the stimulus duration has been changed from '1250 ms' to '1000–1500 ms'. This deviation from Stage 1 received editorial approval and was corrected before data collection.

physiological RTT (very difficult; table 3). Each increase in the difficulty level was indicated by specific sounds and a visual animation, and the awarded points were multiplied.

A limited number of errors were allowed per run, as indicated by both a speed and an accuracy gauge with five levels each. When the participant responded too late to a Go trial, the speed gauge loosed one level. When the participant responded to a NoGo trial or did not respond to a Go trial, the accuracy gauge loosed one level. This two 'life' gauges system was implemented to encourage participants to remain in a stable speed–accuracy trade-off. As soon as one of the two gauges was emptied, the game was over. At the end of a run, a score screen appeared informing the participants about their gain in in-game money, and their local ranking (i.e. how well they performed based on their own previous scores). If the last score was the participant's best score or if he/she reached his/her highest difficulty level, a rewarding animation was played.

The rewarded in-game currency allowed buying 'power-ups': a 'life' regeneration filling the two gauges of one level every 20 s, a total 'life' increase providing two supplementary levels to both gauges at the start of the game, a temporary score multiplier power and a possibility to decrease the difficulty of two levels at any moment. Currency was task-specific, and to buy power-ups for a specific task, you needed currencies from the other task. As such, if participants wanted to reach a high score at a given task, they had to play as much at the other task. A detailed description of the intervention gamification mechanisms is available as supplementary material in the study OSF page.

A global ranking table (i.e. the participant's best scores compared with the other participants) could be accessed from the start menu. Fake high and low scores were implemented in our database to motivate the participants while avoiding discouraging the less successful ones.

The daily requirement of training time for each task was represented by gauges in the level selection screen, where one gauge corresponded to 10 min of training time. After each training session, the corresponding gauge increased based on the amount played. Once the gauge for a given day was full, participants should have stopped their daily training for the given task. To complete the intervention, they needed to fill 20 gauges for each task in four weeks (total of 400 min).

### 2.7.1. Gamifications

State-of-the-art principles of general game design have guided the whole design of the intervention to create a satisfying experience for video-game players and to reinforce their intrinsic motivation [56,57]. Namely, we notably included:

— Clear intrinsic and social challenging goals with progressive difficulty levels and a feedback on participants' ranking against their own and other players' scores. Since points are poorly interesting without internal value, the social comparison increases the perception of the reward for the participant and increases their importance. We also propose different gameplay mechanics, allowing participants to take a rest by switching among games and varying their overall experience, thereby preventing them from stopping playing the game because of boredom or frustration.

**Table 3.** Difficulty parameters at each level for all tasks (in seconds).

| | 1 | 2 | 3 | 4 | 5 | 6 | 7 | 8 | 9 | 10 | 11 | 12 | 13 | 14 | 15 | 16 | 17 | 18 |
|---|---|---|---|---|---|---|---|---|---|---|---|---|---|---|---|---|---|---|
| GNG (RTT) | 1.1 | 1 | 0.9 | 0.8 | 0.725 | 0.675 | 0.625 | 0.575 | 0.55 | 0.525 | 0.5 | 0.475 | 0.4525 | 0.43 | 0.407 | 0.387 | 0.36 | 0.33 |
| CAT (1.25–GSD) | 0.88 | 0.81 | 0.74 | 0.67 | 0.62 | 0.57 | 0.53 | 0.49 | 0.455 | 0.42 | 0.39 | 0.36 | 0.335 | 0.31 | 0.29 | 0.27 | 0.26 | 0.25 |

— An in-game currency mechanic to motivate the participants to perform better and to allow them to reiterate the same experience but with new game configurations after they acquired power-ups. The currency also creates internal values: the scoring/cash mechanisms are strongly rewarding for the players because they can access additional contents after few hours of play. This possibility enables new actions that renew the gaming experience and allows players to develop new complex strategies after they have mastered the basic mechanics.

— An artistic direction of the gaming environment to create a rich and joyful experience for the players. From aesthetics point of view, each mechanic has a specific visual design, with small cinematics making the universe alive; the music is procedurally generated by mixing chunks and it helps to avoid boredom; multiple sound effects support the overall experience and give direct feedback to the players about their performance. From reward and punishment points of view, the art direction has focused on sound effects, which are more and more pleasant by progressing in the game or shame the player for a mistake, and animations judging the performance for each single action and at the end of the game.

## 2.8. Questionnaires

Demographic information, as well as inclusion and exclusion criteria were assessed with:

— A custom-made 13-item general health questionnaire.
— A custom-made palatability questionnaire: participants had to answer the question 'Imagine eating the mentioned food, how much do you like what you taste ?' on 5-point Likert scale with 'not at all' and 'very much' as anchors to unhealthy salty, unhealthy sweet, healthy salty and healthy sweet items (six items each).

Questions related to the interventions and condition expectation were assessed post-intervention with:

— A French-translated version of the Game Experience Questionnaire core module [52] to assess the entertaining and motivational aspect of the intervention. Twenty-seven items on a 5-point Likert scale with 'not at all' and 'extremely' as anchors concerning the *competence*, *flow*, *tension/annoyance*, *challenge*, *negative affect* and *positive affect* felt by the participant while playing. The items related to the *sensory and imaginative immersion* component were removed as they do not apply to the intervention. A global index for this questionnaire was calculated by grouping the 'competence', 'flow', 'challenge' and 'positive affect' components values with the opposite values of the 'tension/annoyance' and 'negative affect' components.
— A custom-made debriefing post-intervention questionnaire: five items about the feeling and understanding of the intervention's purpose. In this questionnaire, participants' expectations on the effects of the experimental and control interventions are assessed with the following question: 'What do you think the game practice has improved or modified?' If the response includes the notion of better eating habit and/or modification of items ratings (healthy items up and unhealthy items down), the variable 'Expected valuation' took the value 1. If the response includes notion of weight loss, the variable 'Expected weight loss' took the value 1. Any other responses (not understanding the outcome, expectation of weight gain, unhealthy item valuation increase, healthy item valuation decrease, etc.) resulted in the value 0 in their respective variables.

## 2.9. Analytical procedure

ANOVAs were computed using R base function.

Our $\alpha$ threshold was set to 0.05. Partial $\eta^2$ and generalized $\eta^2$ were reported as effect sizes for the ANOVA analyses.

For each distribution, skewness and kurtosis were assessed using the R psych package [58].

In the case of $p$-values above our 0.05 $\alpha$ threshold, Bayes factors were computed using the R BayesFactor package [59] to estimate the likelihood of the null hypothesis.

### 2.9.1. Data exclusion rules

For each distribution (i.e. weight and palatability rating), outlier participants were defined as those outside the 2.5MAD (median average deviation; moderately conservative criterion) range around the median [60] and were removed from the related descriptive and inferential analyses.

Additionally, we planned to remove the entire data of a given participant if he/she meets the following criteria:

— More than 2/20 training days not completed,
— More than 5/20 days with less than 20 min of training,
— If more than a third of the training days (i.e. 7 days) did not exceed the fifth difficulty level (easy difficulty level) at either the GNG or CAT, the entire data of the participant were excluded from the analysis because it would indicate that the participant did not even pay a minimal attention to the tasks during most of the training.[1]

### 2.9.1.1. Palatability ratings

All palatability ratings from a given participant were excluded from the analyses if the pre- or the post-intervention palatability rating questionnaires are not completed.

To ensure a thorough filling of the analogue scales, items with an RT shorter than 300 ms were excluded. Only the trained items (those above the median before the intervention) were considered in the analysis. For each participant, the mean rating was calculated after trimming the 20% highest and the 20% lowest rated items at pre-intervention from the healthy and unhealthy item distribution to only target the items having the room to change, thus preventing ceiling and floor effect.

### 2.9.1.2. Weight

Weight data were removed if the participant indicates not having followed the measurement guideline (i.e. weighing just after waking up and going to the toilets).

Changes in weight were assessed using three deltas: pre- versus post-intervention; post-intervention versus 1-month follow-up; 1 versus 4-month follow-up. For each distribution of these deltas, outlier changes in weight values were defined as those outside the 3MAD range around the median, and the concerned data points were excluded from the analyses.

All weight data from a given participant were excluded from the analyses if at least one of his/her data points is missing. In the case where more than 20% of all participants had missed at least one weight data point, every available weight data point would have been kept, and another statistical design would have been used to prevent losing too much statistical power (see the Statistical contrasts section).

### 2.9.2. Statistical contrasts and predictions

H1: *Modification in palatability rating.* The modification in participants' palatability ratings were assessed by the triple interaction term of an Intervention (Experimental versus Control) × Session (Pre-, Post-intervention) × Food type (Healthy; Unhealthy) mixed ANOVA.

H1: a larger increase in the difference between unhealthy than healthy items' palatability ratings in the experimental than control group between the pre- and post-intervention.

As planned contrasts, this interaction was then split into two Intervention × Session mixed ANOVAs with the unhealthy palatability rating and the healthy palatability as dependent variables.

— H1a: a larger decrease in unhealthy items' palatability ratings in the experimental than the control group between the pre- and post-intervention;
— H1b: the reverse pattern for the healthy items palatability ratings.

H2: *Weight loss.* The changes on participants' weight were assessed by the interaction term of an Intervention (Experimental versus Control) × Session (Pre-, Post- intervention, 1- and 4-Month follow-up) mixed ANOVA. If more than 20% of our participants were excluded (see the Data exclusion rules section), the analyses of the weight would have been separated in three contrasts: pre-intervention versus post-intervention, pre-intervention versus 1 month and pre-intervention versus 4 months.

We predict a larger decrease in weight in the experimental than in the control group.

---

[1]The goal of this criterion was to detect and exclude participants not engaged in the training tasks. This has been modified from the Stage 1 manuscript because the initial criterion did not allow to control for this aspect: we initially used the linear slope of performance throughout the intervention as the engagement criterion, which is a poor index of engagement in the training since one can be engaged in the training without improving due to, for example, ceiling effect. This deviation has received editorial approval and was corrected before data analysis.

H3: *Changes in palatability ratings mediate changes in weight*. If the ANOVA interaction on weight reaches significance and is driven by a larger weight loss for the experimental than control group, a causal mediation analysis with 1000 bootstraps (comparable to Lawrence *et al.* [35]) would be performed using the R mediation package [61], with the Intervention factor as independent variable, the largest significant weight loss across all time-points comparisons as dependent variable and changes in Food Type rating (i.e. mean unhealthy rating/mean healthy rating) as mediator. If the average causal mediation effect (i.e. the indirect effect) was significant, then changes in healthy and/or unhealthy palatability ratings would have been interpreted as mediating the effect of the intervention on weight loss.

### 2.9.3. Questionnaires

For the Game Experience Questionnaire, only descriptive violin plot for all six components' scores were reported as a way to describe the entertaining and motivational value of the game.

## 2.10. Positive controls and quality checks[2]

### 2.10.1. Between-group baselines

To verify that the intervention condition assignment resulted in properly balanced baselines, we assessed whether we observed differences of Cohen's $d < 0.3$ between the experimental and control group at pre-intervention on BMI, Age, Gender ratio, trained Unhealthy food rating and trained Healthy food rating.

Since we implemented a selection of the target items based on participants' individual rating and that the assignment was randomized, we did not test with inferential statistics whether the ratings and BMI were equivalent between the two groups, and we thus did not conduct a power analysis for these effects.

### 2.10.2. Primary outcome statistical assumptions

Any biasing of the primary statistical outcomes by ceiling or floor effect were controlled on the palatability ratings by using trimmed means (see the Data exclusion rules section), and on the palatability ratings and weight by using an additional non-parametric statistical approach (Brunner and Langer's non-parametric mixed-effects models [62,63]) in the case of non-normality of the data distributions (see [64] for a discussion).

### 2.10.3. Game experience questionnaire

Only a difference of Cohen's $d < 0.3$ was expected for the global index of the game experience questionnaire between the experimental and control group, as the game aspects are the same between both Interventions. However, if a $d \geq 0.3$ difference was observed, participants with the value the farther from the pooled groups median would have been successively excluded until the groups are balanced.

### 2.10.4. Items liking

A difference of Cohen's $d > 0.3$ was expected between the mean palatability ratings of the trimmed trained items and the middle of the analogue scale (i.e. 50 points). To ensure that the healthy and unhealthy items were actually liked by the participants (and not neutral), if a $d \leq 0.3$ difference was observed, participants with the closest value to 50 from the pooled groups would have been successively excluded until the pooled average differed from 50.

### 2.10.5. Expectancy effect

Expectancy effect was measured based on the count data of both 'Expected valuation' and 'Expected weight loss' binary variables (see the Questionnaires section) on two 2-by-2 contingency matrices, with the $\varphi$ as effect size. In the case of a $\varphi > 0.2$, we would have replaced randomly chosen 'unblinded' and 'blinded' participants from the most divergent count-value until we reach a $\varphi \leq 0.2$.

[2]For the same reason as footnote 3, the positive control comparing performance slopes between groups was removed. This change from Stage 1 was made before data analysis.

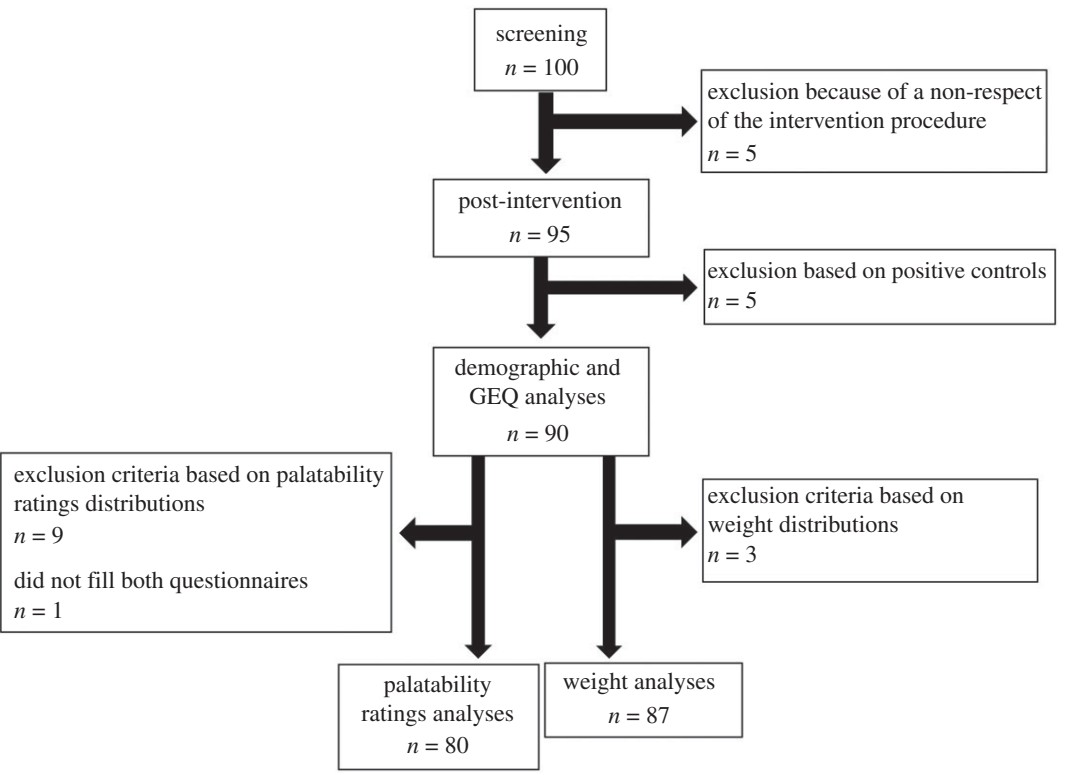

**Figure 4.** Data exclusion from the screening to the different analyses.

**Table 4.** Demographic data.

| mean ± s.d. | control ($n = 44$) | experimental ($n = 46$) |
|---|---|---|
| age | 24.9 ± 4.6 | 25.2 ± 5.7 |
| gender ratio (M/F) | 1.2 (24F/20M) | 1.4 (27F/19M) |
| BMI | 24.5 ± 3.5 | 24.4 ± 3.6 |

For example, in the case where we have eight 'blinded' participants in the control group (35 'unblinded') and one 'blinded' participant in the experimental group (42 'unblinded'), we would replace randomly one of the eight in the control group by a new participant.[3]

The result of the $\chi^2$ test (or Fisher's exact test if a cell has a count data below 5) was reported to better understand the feasibility of this method to create matching expectations. However, we still support brute forcing the matching expectancy to avoid our key contrasts being confounded by this variable.

# 3. Results

## 3.1. Participants

A total of 100 participants were recruited, five of which were excluded during the intervention because they did not comply with the intervention's procedure (i.e. to play 5 days a week for four weeks; see table 4 for demographic data and figure 4 for the data exclusion).

## 3.2. Positive controls

All positive control except the 'Expected Valuation' contingency matrix were respected (see electronic supplementary material). Following our pipeline (see the Positive controls and quality checks

---

[3]The inequality signs were inversed in Stage 1. This change received editorial approval and was made before data collection.

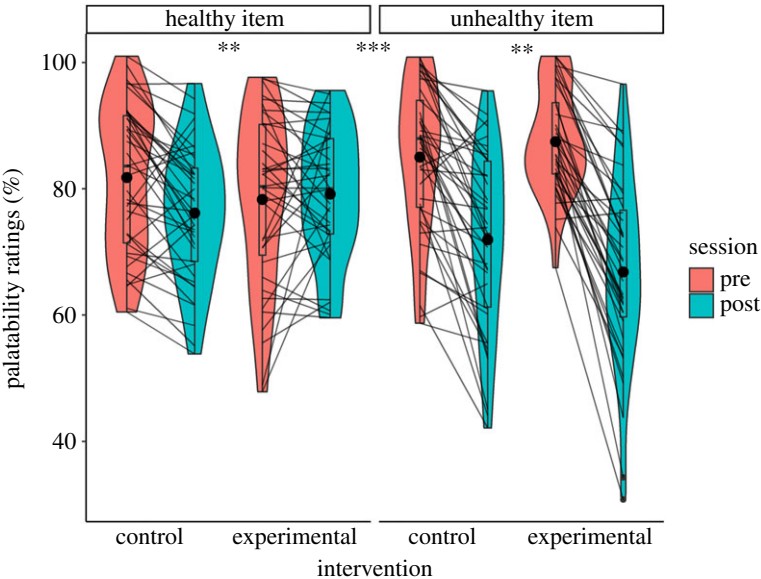

**Figure 5.** Palatability ratings pre- and post-intervention. Trimmed means of palatability ratings are represented for the control and experimental groups. Individual data points, means (bold circle), distributions' density (violin), medians, first and third quartiles (horizontal bars), and the 1.5 inter-quartiles range (whiskers) are represented. $^{**}p < 0.01$, $^{***}p < 0.001$.

**Table 5.** Palatability ratings data.

| mean ± s.d | control ($n = 41$) | | experimental ($n = 39$) | | training × session interaction | training × item × session interaction |
|---|---|---|---|---|---|---|
| | pre-training | post-training | pre-training | post-training | | |
| healthy item rating | 81.8 ± 11.8 | 76.1 ± 10.6 | 78.3 ± 13.8 | 79.2 ± 10.3 | $p = 0.007$ | $p < 0.001$ |
| | | | | | $\eta G^2 = 0.019$ | $\eta G^2 = 0.021$ |
| delta [0.95 CI] | 5.6 [2.9; 8.4] | | −0.9 [−4.8; 3.1] | | | |
| unhealthy item rating | 85 ± 11.9 | 71.9 ± 14 | 87.5 ± 8.3 | 66.8 ± 14.7 | $p = 0.007$ | |
| | | | | | $\eta G^2 = 0.023$ | |
| delta [0.95 CI] | 13.1 [9.2; 17] | | 20.6 [16.8; 24.4] | | | |

sections), we randomly removed participants in the outlier cells until achieving a $\varphi$ below 0.2. A total of five participants had to be removed from all analyses.

## 3.3. Palatability ratings

Detailed results are reported in table 5 and figure 5. All distributions respected our threshold for parametric ANOVAs.

Regarding our exclusion criteria for the palatability ratings analyses: one participant did not complete the post-intervention questionnaire; two were outside the 2.5MAD around the median range of the healthy items' distribution; eight were outside the 2.5MAD around the median range of the unhealthy items' distribution. A total of 10 participants were thus excluded and we reached for this analysis a sample of $n = 80$.

There was a Training by Item by Session triple interaction on the palatability ratings driven by a larger devaluation of unhealthy in the experimental group than the devaluation of the healthy items in the control group ($F_{1,156} = 11.6$, $p < 0.001$, $\eta_G^2 = 0.021$).

For the healthy items, there was a Training by Session interaction driven by a larger items devaluation in the control than in the experimental group ($F_{1,78} = 7.6$, $p = 0.007$, $\eta_G^2 = 0.019$).

For the unhealthy items, there was a Training by Session interaction driven by a larger items devaluation in the experimental than in the control group ($F_{1,78} = 7.7$, $p = 0.007$, $\eta_G^2 = 0.023$).

## 3.4. Weight

Detailed results are reported in table 6 and figure 6. All distributions respected our threshold for parametric ANOVAs.

Regarding our exclusion criteria for the weight analyses: three participants were outside the 2.5MAD around the median range of the weight distribution; seven were outside of the 3MAD around the median range for the pre versus post delta weight; one for the post versus 1 M delta and one for the 1 M versus 4 M delta. However, because 29 participants did not give their weight four months post-intervention (33.3%), the differences between one time point and another were analysed as independent contrasts (see the Statistical contrasts and predictions sections for the decision algorithm), and outlier participants based on the delta distributions were kept. Overall, three participants were excluded for the weight analyses to reach a sample size of $n = 87$ for the pre versus post and the post versus 1 M contrasts, and of $n = 58$ for the 1 versus 4 M contrast.

There was no Training by Session interaction for the pre- versus post-intervention contrast ($F_{1,85} = 0.65$, $p = 0.42$, $\eta_G^2 = 0.000$) supported by a Bayes factors analysis ($BF_{01} = 3.4$), pre-intervention versus one-month contrast ($F_{1,77} = 0.94$, $p = 0.49$, $\eta_G^2 = 0.000$) not supported by a Bayes factors analysis ($BF_{01} = 2.4$) and pre-intervention versus four-month contrast ($F_{1,56} = 2.01$, $p = 0.16$, $\eta_G^2 = 0.000$), not supported by a Bayes factors analysis ($BF_{01} = 1.8$).

Since no contrast showed an effect on weight loss, no mediation analysis between the change in valuation and weight loss was performed.

## 3.5. Game Experience Questionnaire

Detailed results are reported in table 7 and figure 7. The Tension and Negative affects components were reversed, so that a higher score shows a better experience.

Only small differences were observed between the experimental and control groups. The global scores and all components except the Challenge and Flow were above the mean of modalities, indicating a globally good experience of the gamified intervention.

# 4. Discussion

We found that four weeks of combined food-Go/NoGo (GNG) and Cued-Approach Training (CAT) induced a devaluation of participants' favourite unhealthy food cues. The training did not, however, influence the valuation of healthy food cues nor influence the weight of the participants.

## 4.1. Combined Go/NoGo and Cue approach training reduces food cue palatability

We confirmed our hypothesis for a larger decrease in the palatability of the target unhealthy items in the experimental group than control group. We observed a decrease of 21% with a 16.8–24.4 0.95 confidence interval in the experimental group. This effect was 8% above the effect in the control group (13.1; 0.95 CI = [9.2; 17]). Of note, the devaluation effect was robust, with 37 out of the 39 participants (95%) in the experimental group showing a decrease in palatability ratings. Because the effects of expectation, of training adherence and of baseline differences were all strictly controlled, we interpret our effect as following from the biasing of the stimulus–response contingencies. This parameter was indeed the only difference between the experimental and control group. Our results thus confirm that the systematic association of the target items with response inhibition in the GNG task and with the development of attentional bias away to the item in the CAT causally induces the cue devaluation.

As also expected, we observed decreases in the ratings of both healthy and unhealthy trained food items in the control group. We attribute this effect to the control group still having to inhibit responses to these items in the GNG and to ignore them half of the time in the CAT. This pattern suggests a dose–response relationship between the devaluation and the exposure to the stimulus–response association in the GNG and CAT [25].

Furthermore, since 34 out of 44 participants (77%) in the control group reported expecting an increase in the valuation of the healthy items after intervention, expectation is unlikely to account for the

**Table 6.** Weight data.

| mean ± s.d. delta [0.95 CI] | control | | | | experimental | | | | pre versus post-training × session interaction | pre versus 1 M training × session interaction | pre versus 4 M training × session interaction |
|---|---|---|---|---|---|---|---|---|---|---|---|
| | pre-training (n = 42) | post-training (n = 42) | 1 M (n = 26) | 4 M (n = 12) | pre-training (n = 45) | post-training (n = 45) | 1 M (n = 27) | 4 M (n = 13) | | | |
| weight | 70 ± 11.1 | 69.6 ± 11.3 | 69.6 ± 10.8 | 67.5 ± 10.4 | 72 ± 11 | 71.8 ± 11 | 70.4 ± 9.9 | 71.6 ± 10.3 | $p = 0.42$ | $p = 0.34$ | $p = 0.16$ |
| | | | | | | | | | $\eta_G^2 = 0.000$ | $\eta_G^2 = 0.000$ | $\eta_G^2 = 0.000$ |
| | | | | | | | | | $BF_{01} = 3.4$ | $BF_{01} = 2.4$ | $BF_{01} = 1.8$ |

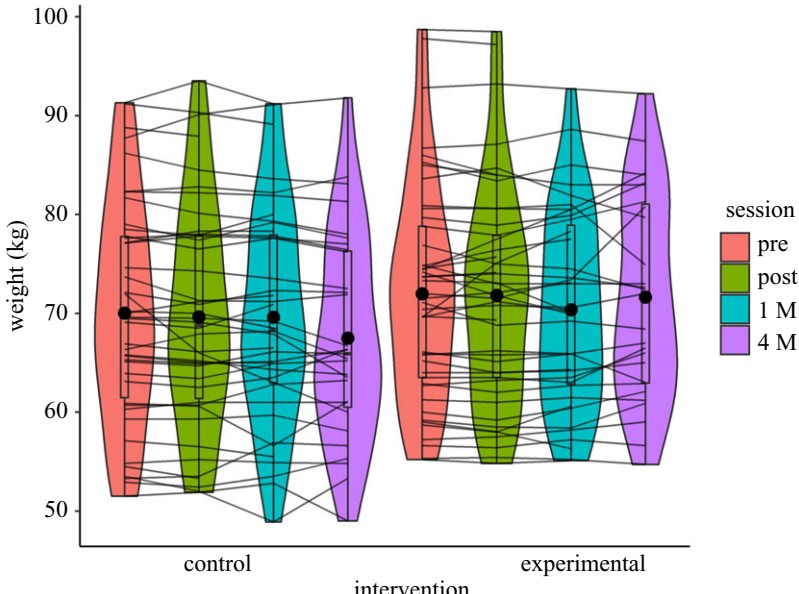

**Figure 6.** Weight in kilogram at pre- and post-intervention, and one and four months post-intervention. Both the control and experimental groups are represented for all trimmed palatability means. Individual data points, means (bold circle), distributions' density (violin), medians, first and third quartiles (horizontal bars), and the 1.5 inter-quartiles range (whiskers) are represented. No effects reached significance.

**Table 7.** Game Experience Questionnaire scores.

| mean ± s.d. | global score | competence | flow | tension (reversed score) | challenge | negative affects (reversed score) | positive affects |
|---|---|---|---|---|---|---|---|
| control (n = 44) | 56.2 ± 10.82 | 9.4 ± 3.9 | 8 ± 3.6 | 8.8 ± 2.5 | 8.1 ± 3.3 | 10.9 ± 2.8 | 11.1 ± 3.6 |
| experimental (n = 46) | 54.8 ± 9.8 | 10.3 ± 3.4 | 7.5 ± 3.9 | 9.4 ± 2.4 | 7 ± 3.4 | 10.8 ± 2.9 | 9.7 ± 3.5 |

devaluation of healthy food items. By contrast, the devaluation of unhealthy food items may have followed from both the effect of expectation and of the intervention. Hence, the difference in effect size between the devaluation of healthy and unhealthy food items in the control group might follow from the effects of expectation. We thus conclude from our data that the expectation for a devaluation of unhealthy items with the intervention accounts for 57% of the observed decrease in unhealthy items' ratings in the control group.

We expected the response to Go items in the CAT and in the GNG to increase the perceived value of the healthy items in the experimental group compared with the control group. Our results, however, do not confirm this hypothesis. This null result can be explained by either our measure of palatability rating not being sensitive to the effect of CAT or by the way we implemented the CAT in our intervention. The CAT has proven efficient to change monetary value [8], food preference during choice task [8–10] and food intake [11], but not on food palatability rating as in the present study. The CAT might have improved the trained healthy items' preference over the unhealthy items without changing how participants rate their palatability based on pictures. Regarding the specificities of our CAT, the main difference with previous implementation lies in the presence of feedback on performance. In contrast with our CAT, previous CAT did not include any external reinforcers, even though internal reinforcement probably occurred since participants were aware of their performance.

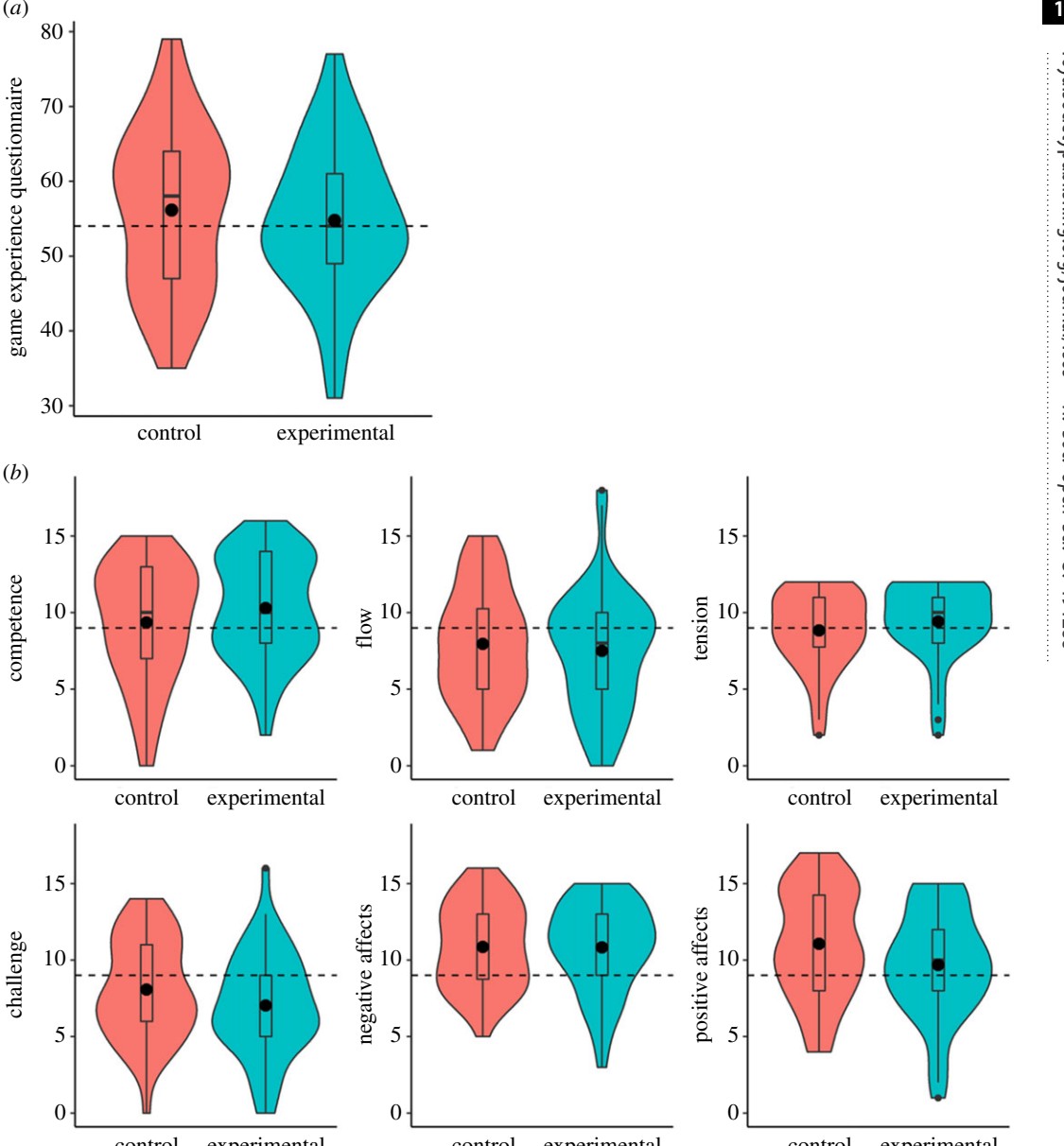

**Figure 7.** (*a*) Game Experience Questionnaire global score. (*b*) Game Experience Questionnaire components. Both control and experimental groups are represented for all scores. Means (bold circle), distributions' density (violin), medians, first and third quartiles (horizontal bars), and the 1.5 inter-quartiles range (whiskers) are represented. The modalities' average is represented by dashed horizontal lines. Both tension and negative affects components are reversed, with a higher score representing a better experience.

## 4.2. Reduction in food cue palatability is not associated with weight loss

As a second outcome, we assessed whether the training influenced participants' weight, with the idea that if decreases in unhealthy food palatability rating induce a healthier diet, weight loss may occur. This measure also has the advantage to be easy to collect in an online protocol while having a good reliability [27,35,36,65,66].

We, however, found no effect of the training on participants' weight. This finding may appear surprising given previous evidence for a positive association between food palatability and consumption [67,68]. Possible accounts for our finding include the following. First, participants weighed themselves at home with their own scales, which may have increased the variance and thus decreased our power to detect true effects. Participants were, however, thoroughly instructed to weight themselves directly after waking up, before eating and after going to the toilets, reducing the

natural weight variance. The design was also within-subject and thus even if the participants' scales were biased, differences in their weights should have still been detected. Second, our training may not have been able to induce detectable change in weight in our population with healthy baseline weights. Weight loss induced by food-ICT on healthy population similar to ours were, however, detected in previous studies [36,65,66]. In addition, we recruited participants having a minimum BMI of 20 during screening, without an upper limit. The baseline BMI of the experimental group was at the upper limit of a healthy standard (mean = 24.4, 0.95CI [23.3; 25.5]), with 16 out of 46 participants (35%) in the experimental group defined as overweight (i.e. BMI > 25). Our group thus had a potential for weight loss. Third, it might be that the participants did not consume the trained product frequently, even if they rated them as highly palatable. This account is unlikely because one inclusion criterion was to not follow any restrictive diet; the participants thus tended to consume what they liked the most. However, the participants may not have been fully responsible for the food they consume, as, for example, if they live with their parents or friends. Overall, we conclude that the intervention did not have any, or a sufficiently large effect on weight.

## 4.3. Task gamification ensured a good adherence to the intervention

The core module of the Game Experience Questionnaire [52] revealed that most of participants enjoyed the tasks. Only five participants out of 100 were excluded for not respecting the intervention's procedure (i.e. playing 20 min 5 days a week for four weeks) or not reaching our minimal adherence threshold. Moreover, participants used on average 30 'power-ups' during the training (0.95CI [27; 33], i.e. bonus acquirable during training with in-game currency, with only one participant out of the 95 not using any power-up. Some participants also orally reported playing more than what was expected to increase their position in the global ranking. While we did not contrast this pattern with a control group using a non-gamified version of the task, these results suggest that the gamification created a highly engaging and motivational environment.

# 5. Conclusion

One month of combined GNG and CAT practice induces a robust decrease in unhealthy items' palatability ratings, with no influence on participants' weight. Since expectations and baseline group differences were strictly controlled, only the repeated inhibition of motor responses to the item and the development of attentional bias away to the item account for our effect. We further demonstrate that gamification helps engagement and adherence to the training, facilitating long-term online intervention.

Ethics. All procedures were approved by the Commission cantonale (VD) d'ethique de la recherche sur l'etre humain (CER-VD; protocol 2019-02149).

Data accessibility. All documents related to the intervention programme are available for download on the study OSF page (https://osf.io/4a7wh/?view_only=1bd7aacf997c4441815bbf231233db43). The data are provided in electronic supplementary material [69].

Authors' contributions. L.S. conceptualized the study; L.S. and H.N. designed the study and wrote the manuscript; H.N. collected and analysed the data with the participation of M.M.; M.R. developed the software with the participation of the other authors; M.R. and M.M. reviewed the manuscript. All authors gave final approval for publication.

Competing interests. H.N., M.R., M.M. and L.S. are co-founders and shareholders of Neuria, a company that produces therapeutic video games. The Diner, the app used in the present study, is commercialized by Neuria.

Funding. This work was supported by a grant from the Swiss National Science Foundation (grant no. 320030_175469 to L.S.).

Acknowledgements. We thank Eva Rohrbach for her help in setting up the experiment, and Lea Hartmann and Clare Baragiola from the Metabolic Center of the Fribourg Cantonal Hospital for advice on the nutritional value of the food items.

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
