## [Peer Review File · Royal Society Open Science]

Review History

RSOS-191288.R1 (Original submission)

Review form: Reviewer 1

Is the language acceptable?

Yes

Do you have any ethical concerns with this paper?

No

Have you any concerns about statistical analyses in this paper?

No

Recommendation?

Major revision

Comments to the Author(s)

All comments can be found within the attached document (see Appendix A).

Review form: Reviewer 2

Is the language acceptable?

Yes

Do you have any ethical concerns with this paper?

No

Have you any concerns about statistical analyses in this paper?

No

Recommendation?

Major revision

Comments to the Author(s)

Dear editor,

Below is my review of the manuscript by Najberg and colleagues. Let me start by stating that the concept of registered reports makes great sense and I fully endorse this way of conducting research. I think the topic of the research is timely, and connects greatly to a recent recommendation to perform more well-powered preregistered experiments in the domain of ICT training (Carbine and Larson, 2019). Therefore, think the work lends itself really well for a registered report. I do have a number of comments that the authors may want to address before conducting the experiment. I have listed a number of issues and recommendations of how to improve the proposed research (at least in my opinion) below.

First, I think the decision to include the stop signal task requires more elaboration, and it may perhaps be wise to remove this training procedure from the intervention. The go/no-go task and cue-approach training have shown robust effects on food choice lasting for weeks to months (Chen et al., 2019; JPSP; Salomon et al., 2018; Scientific Reports), and food evaluation (e.g., Chen et al., 2016; JEP:GEN; Quandt et al., 2019; JEP:HPP) as suggested by well-powered preregistered experiments. However, as I understand it, it is less clear whether the stop signal training leads to behavior change, or changes in food evaluation, and there are some slight indications in the literature that this training might not work as well (Jones et al., 2016; Appetite; Veling et al., 2017; Curr. Add. Reports). That is because during stop signal training participants often erroneously respond to the stop items (due to the staircase procedure of presenting the stop signals late during a trial) which may counteract effects of the other training procedures (not responding to no-go items). Note that this is different from cue approach training and go/no-go training where participants never respond to the no-go (or uncued) items.

Second, I think I do not quite understand the power analyses, as I was surprised by the modest proposed sample size. Recent work on cue-approach training and go/no-go training reporting power analyses suggest that in order to find effects on food choice in single session experiments employing a within-subjects design using repeated measurements of food choice between go (or cued) and no-go (or uncued) items 25 or more participants are required (Solomon et al., 2018; Chen et al., 2019). Moreover, effects of go/no-go training on food evaluation in a within design requires 30 participants to arrive at a power of .80 (e.g., Chen et al., 2016). Here the authors want to compare two between-subjects conditions, observe effects after some delay, and do not have any within-subjects baseline measures for the palatability ratings (as I understand it; see also my comment on how this may be addressed). Based on these recent findings, I cannot imagine 25 participants per group will be sufficient to detect a decrease in evaluations between (rather than within) the groups.

As I understand cue-approach training, it is a training to boost attention to go (or cued) items. The training does not aim to reduce responses to no-go (uncued) items. Empirically, all cue-approach training experiments that I have seen always compare go (cued) items with no-go (uncued items) leaving unanswered the question whether the training increases responses to go items or decreases responses to no-go items. In contrast, effects of go/no-go training on palatability ratings have been evaluated against a neutral (untrained) baseline, and effects appear to be driven by decreased evaluations of no-go items, except when a staircase on go trials is used in which evaluations of go items may go up (Chen et al., 2016; Experiment 2). Based on these findings and theoretical grounds, I would therefore predict that evaluations for go healthy foods would go up (as a result of cue approach training and staircase on go trials in go no-go task) and evaluations of no-go foods would go down (as a result of go/no-go training). Have the authors considered this possibility? The introduction may also benefit from discussing possible different working mechanisms of the different training procedures.

Based on the arguments above I think it might be good to employ untrained baseline items in the design for both go healthy foods and no-go unhealthy foods. This may not only increase sensitivity to detect effects, but also aid with interpretation of the data on whether effects are more likely driven by increased ratings of healthy food items, or decreased ratings of unhealthy items. One way of doing this would be to apply matched (based on initial ratings) randomization of the food items into trained and untrained healthy food items and trained and untrained unhealthy foods. The statistical design would then be intervention (exp vs control) x session x item type (untrained trained) x food type (healthy vs unhealthy) where the last three factors are within-participants factors. It would require sufficient items per within-subject condition.

Have the authors also considered the possibility of including a series of binary choices (e.g., as in Schonberg et al., 2014; Salomon et al., 2018; Chen et al., 2019) between go and no-go items pre and post? Work suggests both cue-approach training and go/no-go training show robust effects on such measurements (see citations above), and there is even one experiment showing that the probability of choosing healthy over unhealthy can be increased by training (Chen et al., 2019). Here it would also be possible to include baseline (untrained) choice pairs.

It would be better if it is possible to measure weight objectively, but perhaps that is not possible. A choice measure as described above might be another good measure of behavior.

Gamifying training tasks might be risky because subtle changes may influence effects. For instance, by changing some of the task parameters (e.g., employing a staircase in go / no-go training) attention may shift toward certain item types (e.g., more attention to go items than no-go items) which may influence palatability ratings in some way (e.g., leading to stronger go valuation instead of no-go devaluation; see e.g., Quandt et al., 2019; JEP:HPP). In light of this I wondered whether the authors have considered to pilot the gamified intervention in a single session experiment to be at least sure the task works on the short-term to influence palatability ratings (my experience is that go/no-go virtually always works to decrease evaluations of no-go versus go items from pre to post when the evaluations are matched on the pre-rating). Perhaps it is possible to test this first before offering these tasks for a more intensive training especially in light of the fact that the new gamified tasks are slightly modified from the original tasks.

I am not sure about removing palatability ratings slower than 1500 (this may be needed for choices but not ratings; for a discussion see Chen et al., 2019).

Maybe I missed it, but do the authors also measure possible differences in expectations between the intervention and control group? This seems important because this is one of the major limitations they seem to address.

I hope my comments and suggestions are useful to further improve the proposed research. I look forward to learn about the final experimental protocol and results.

Decision letter (RSOS-191288.R0)

10-Sep-2019

Dear Dr Spierer,

The Editors assigned to your Stage 1 Registered Report ("Decreasing unhealthy food items valuation and weight with gamified executive control training") have now received comments from reviewers. We would like you to revise your paper in accordance with the referee and editors suggestions which can be found below (not including confidential reports to the Editor). Please note this decision does not guarantee eventual acceptance.

When submitting your revised manuscript, you must respond to the comments made by the referees and upload a file "Response to Referees" in "Section 2 - File Upload". Please use this to document how you have responded to the comments, and the adjustments you have made. In order to expedite the processing of the revised manuscript, please be as specific as possible in your response.

Kind regards,
Professor Chris Chambers
Royal Society Open Science
openscience@royalsociety.org

Associate Editor Comments to Author (Professor Chris Chambers):

Associate Editor: 1

Comments to the Author:

Two expert reviewers have now appraised the manuscript and provided very high quality reviews. Both reviewers are cautiously positive about the proposal but also raise a series of major concerns that cut across the full range of the Stage 1 review criteria, including the rationale for the research question (and choice of training interventions -- a point highlighted by both reviewers), level of methodological detail, justification of sample size (and power analysis), and robustness of the study design (including positive controls). All of these issues will need to comprehensively addressed to achieve IPA.

Comments to Author:

Reviewer: 1

Comments to the Author(s)

All comments can be found within the attached document.

Reviewer: 2

Comments to the Author(s)

Dear editor,

Below is my review of the manuscript by Najberg and colleagues. Let me start by stating that the concept of registered reports makes great sense and I fully endorse this way of conducting research. I think the topic of the research is timely, and connects greatly to a recent recommendation to perform more well-powered preregistered experiments in the domain of ICT training (Carbine and Larson, 2019). Therefore, think the work lends itself really well for a registered report. I do have a number of comments that the authors may want to address before conducting the experiment. I have listed a number of issues and recommendations of how to improve the proposed research (at least in my opinion) below.

First, I think the decision to include the stop signal task requires more elaboration, and it may perhaps be wise to remove this training procedure from the intervention. The go/no-go task and cue-approach training have shown robust effects on food choice lasting for weeks to months (Chen et al., 2019; JPSP; Salomon et al., 2018; Scientific Reports), and food evaluation (e.g., Chen et al., 2016; JEP:GEN; Quandt et al., 2019; JEP:HPP) as suggested by well-powered preregistered experiments. However, as I understand it, it is less clear whether the stop signal training leads to behavior change, or changes in food evaluation, and there are some slight indications in the literature that this training might not work as well (Jones et al., 2016; Appetite; Veling et al., 2017; Curr. Add. Reports). That is because during stop signal training participants often erroneously respond to the stop items (due to the staircase procedure of presenting the stop signals late during a trial) which may counteract effects of the other training procedures (not responding to no-go items). Note that this is different from cue approach training and go/no-go training where participants never respond to the no-go (or uncued) items.

Second, I think I do not quite understand the power analyses, as I was surprised by the modest proposed sample size. Recent work on cue-approach training and go/no-go training reporting power analyses suggest that in order to find effects on food choice in single session experiments employing a within-subjects design using repeated measurements of food choice between go (or cued) and no-go (or uncued) items 25 or more participants are required (Solomon et al., 2018; Chen et al., 2019). Moreover, effects of go/no-go training on food evaluation in a within design requires 30 participants to arrive at a power of .80 (e.g., Chen et al., 2016). Here the authors want to compare two between-subjects conditions, observe effects after some delay, and do not have any within-subjects baseline measures for the palatability ratings (as I understand it; see also my comment on how this may be addressed). Based on these recent findings, I cannot imagine 25 participants per group will be sufficient to detect a decrease in evaluations between (rather than within) the groups.

As I understand cue-approach training, it is a training to boost attention to go (or cued) items. The training does not aim to reduce responses to no-go (uncued) items. Empirically, all cue-approach training experiments that I have seen always compare go (cued) items with no-go (uncued items) leaving unanswered the question whether the training increases responses to go items or decreases responses to no-go items. In contrast, effects of go/no-go training on palatability ratings have been evaluated against a neutral (untrained) baseline, and effects appear to be driven by decreased evaluations of no-go items, except when a staircase on go trials is used in which evaluations of go items may go up (Chen et al., 2016; Experiment 2). Based on these findings and theoretical grounds, I would therefore predict that evaluations for go healthy foods would go up (as a result of cue approach training and staircase on go trials in go no-go task) and evaluations of no-go foods would go down (as a result of go/no-go training). Have the authors

considered this possibility? The introduction may also benefit from discussing possible different working mechanisms of the different training procedures.

Based on the arguments above I think it might be good to employ untrained baseline items in the design for both go healthy foods and no-go unhealthy foods. This may not only increase sensitivity to detect effects, but also aid with interpretation of the data on whether effects are more likely driven by increased ratings of healthy food items, or decreased ratings of unhealthy items. One way of doing this would be to apply matched (based on initial ratings) randomization of the food items into trained and untrained healthy food items and trained and untrained unhealthy foods. The statistical design would then be intervention (exp vs control) x session x item type (untrained trained) x food type (healthy vs unhealthy) where the last three factors are within-participants factors. It would require sufficient items per within-subject condition.

Have the authors also considered the possibility of including a series of binary choices (e.g., as in Schonberg et al., 2014; Salomon et al., 2018; Chen et al., 2019) between go and no-go items pre and post? Work suggests both cue-approach training and go/no-go training show robust effects on such measurements (see citations above), and there is even one experiment showing that the probability of choosing healthy over unhealthy can be increased by training (Chen et al., 2019). Here it would also be possible to include baseline (untrained) choice pairs.

It would be better if it is possible to measure weight objectively, but perhaps that is not possible. A choice measure as described above might be another good measure of behavior.

Gamifying training tasks might be risky because subtle changes may influence effects. For instance, by changing some of the task parameters (e.g., employing a staircase in go / no-go training) attention may shift toward certain item types (e.g., more attention to go items than no-go items) which may influence palatability ratings in some way (e.g., leading to stronger go valuation instead of no-go devaluation; see e.g., Quandt et al., 2019; JEP:HPP). In light of this I wondered whether the authors have considered to pilot the gamified intervention in a single session experiment to be at least sure the task works on the short-term to influence palatability ratings (my experience is that go/no-go virtually always works to decrease evaluations of no-go versus go items from pre to post when the evaluations are matched on the pre-rating). Perhaps it is possible to test this first before offering these tasks for a more intensive training especially in light of the fact that the new gamified tasks are slightly modified from the original tasks.

I am not sure about removing palatability ratings slower than 1500 (this may be needed for choices but not ratings; for a discussion see Chen et al., 2019).

Maybe I missed it, but do the authors also measure possible differences in expectations between the intervention and control group? This seems important because this is one of the major limitations they seem to address.

I hope my comments and suggestions are useful to further improve the proposed research. I look forward to learn about the final experimental protocol and results.

Author's Response to Decision Letter for (RSOS-191288.R0)

See Appendix B.

RSOS-191288.R1 (Revision)

Review form: Reviewer 1

Do you have any ethical concerns with this paper?

No

Recommendation?

Accept with minor revision

Comments to the Author(s)

Please see attached document (Appendix C).

Review form: Reviewer 2

Do you have any ethical concerns with this paper?

No

Recommendation?

Accept in principle

Comments to the Author(s)

Dear Editor,

I think the authors addressed my previous comments adequately. I would do some things differently, such as employ an objective measure of behavior, or employ within-subjects baselines, but I can also see why the authors choose not to do this. I do not agree with the authors that their measure of weight is objective as the participants are asked to report it to the experimenters, which makes it self-report. However, if it is not possible to weigh them in the lab, then so be it. Therefore, I recommend the paper for publication.

Decision letter (RSOS-191288.R1)

28-Oct-2019

Dear Dr Spierer,

On behalf of the Editors, I am pleased to inform you that your Stage 1 Registered Report RSOS-191288.R1 entitled "Modifying food items valuation and weight with gamified executive control training" has been accepted in principle for publication in Royal Society Open Science subject to minor revision in accordance with the referee and editor suggestions. Please find their comments at the end of this email.

To revise your manuscript, log into <https://mc.manuscriptcentral.com/rsos> and enter your Author Centre, where you will find your manuscript title listed under "Manuscripts with Decisions". Under "Actions," click on "Create a Revision." You will be unable to make your

revisions on the originally submitted version of the manuscript. Instead, revise your manuscript and upload a new version through your Author Centre.

When submitting your revised manuscript, you will be able to respond to the comments made by the referees and you should upload a file "Response to Referees". You can use this to document any changes you make to the original manuscript. In order to expedite the processing of the revised manuscript, please be as specific as possible in your response to the referees.

Full author guidelines can be found here <https://royalsocietypublishing.org/rsos/registered-reports>.

Kind regards,
Lianne Parkhouse
Editorial Coordinator
Royal Society Open Science
openscience@royalsociety.org

on behalf of Professor Chris Chambers (Subject Editor, Royal Society Open Science)
openscience@royalsociety.org

Associate Editor Comments to Author (Professor Chris Chambers):

The two original expert reviewers have assessed the revised manuscript. Reviewer 2 is now broadly satisfied and recommends IPA (though the authors should take into account the reviewer's point of disagreement), whereas Reviewer 1 notes a number of remaining ambiguities in the rationale and hypotheses where clarification is needed. A minor revision is therefore invited to address these remaining points.

Reviewer comments to Author:

Reviewer: 1
Comments to the Author(s)
Please see attached document.

Reviewer: 2
Comments to the Author(s)

I think the authors addressed my previous comments adequately. I would do some things differently, such as employ an objective measure of behavior, or employ within-subjects baselines, but I can also see why the authors choose not to do this. I do not agree with the authors that their measure of weight is objective as the participants are asked to report it to the experimenters, which makes it self-report. However, if it is not possible to weigh them in the lab, then so be it. Therefore, I recommend the paper for publication.

Author's Response to Decision Letter for (RSOS-191288.R1)

See Appendix D.

Decision letter (RSOS-191288.R2)

14-Nov-2019

Dear Dr Spierer

On behalf of the Editor, I am pleased to inform you that your Manuscript RSOS-191288.R2 entitled "Modifying food items valuation and weight with gamified executive control training" has been accepted in principle for publication in Royal Society Open Science. The reviewers' and editors' comments are included at the end of this email.

You may now progress to Stage 2 and complete the study as approved. Before commencing data collection we ask that you:

- 1) Update the journal office as to the anticipated completion date of your study.
- 2) Register your approved protocol on the Open Science Framework (<https://osf.io/>) or other recognised repository, either publicly or privately under embargo until submission of the Stage 2 manuscript. Please note that a time-stamped, independent registration of the protocol is mandatory under journal policy, and manuscripts that do not conform to this requirement cannot be considered at Stage 2. The protocol should be registered unchanged from its current approved state, with the time-stamp preceding implementation of the approved study design.

Following completion of your study, we invite you to resubmit your paper for peer review as a Stage 2 Registered Report. Please note that your manuscript can still be rejected for publication at Stage 2 if the Editors consider any of the following conditions to be met:

- The results were unable to test the authors' proposed hypotheses by failing to meet the approved outcome-neutral criteria.
- The authors altered the Introduction, rationale, or hypotheses, as approved in the Stage 1 submission.
- The authors failed to adhere closely to the registered experimental procedures. Please note that any deviations from the approved experimental procedures must be communicated to the editor immediately for approval, and prior to the completion of data collection. Failure to do so can result in revocation of in-principle acceptance and rejection at Stage 2 (see complete guidelines for further information).
- Any post-hoc (unregistered) analyses were either unjustified, insufficiently caveated, or overly dominant in shaping the authors' conclusions.
- The authors' conclusions were not justified given the data obtained.

We encourage you to read the complete guidelines for authors concerning Stage 2 submissions at <https://royalsocietypublishing.org/rsos/registered-reports#ReviewerGuideRegRep>. Please especially note the requirements for data sharing, reporting the URL of the independently registered protocol, and that withdrawing your manuscript will result in publication of a Withdrawn Registration.

Please note that Royal Society Open Science will introduce article processing charges for all new submissions received from 1 January 2018. Registered Reports submitted and accepted after this date will ONLY be subject to a charge if they subsequently progress to and are accepted as Stage 2 Registered Reports. If your manuscript is submitted and accepted for publication after 1 January 2018 (i.e. as a full Stage 2 Registered Report), you will be asked to pay the article processing charge, unless you request a waiver and this is approved by Royal Society Publishing.

You can find out more about the charges at <https://royalsocietypublishing.org/rsos/charges>. Should you have any queries, please contact openscience@royalsociety.org.

Once again, thank you for submitting your manuscript to Royal Society Open Science and we look forward to receiving your Stage 2 submission. If you have any questions at all, please do not hesitate to get in touch. We look forward to hearing from you shortly with the anticipated submission date for your stage two manuscript.

on behalf of Professor Chris Chambers (Registered Reports Editor, Royal Society Open Science)
openscience@royalsociety.org

Author's Response to Decision Letter for (RSOS-191288.R2)

See Appendix E.

RSOS-191288.R3 (Revision)

Review form: Reviewer 2

Is the manuscript scientifically sound in its present form?

Yes

Are the interpretations and conclusions justified by the results?

Yes

Is the language acceptable?

Yes

Do you have any ethical concerns with this paper?

No

Have you any concerns about statistical analyses in this paper?

No

Recommendation?

Accept with minor revision

Comments to the Author(s)

Review RSOS

Dear Editor,

I only have a few comments, as this is a stage 2 registered report. I do not have the time to check their data analyses or perform a detailed comparison of documents. I trust the authors they did this adequately.

I think the most important question raised by the findings how effects can be found on value and not on weight. This raises a number of questions. First, are palatability ratings predictive for consumption? This seems logical but has this been tested before? Second, one explanation for why effects on weight are absent is that participants do not consume the trained products frequently. Can this be ruled out? This could parsimoniously explain why effects are found on value but not on weight. Another explanation could be that participants are not always responsible for the products and amount of food they consume as they may live together with others that are responsible for this. Can this explain the results?

I did not understand the description of the go/no-go task. The healthy foods are not mentioned in this description but are quite crucial. Moreover the difference between the experimental and control conditions is not explained. It is only understandable by looking at the tables.

Minor points:

Results for weight are not mentioned in the abstract. Moreover, it would be good to make explicit in the abstract how many hypothesis were tested and which were confirmed.

There are a number of paragraphs in the intro that consist of only one sentence.

Page 20 line 8; followed

Decision letter (RSOS-191288.R3)

Dear Dr Spierer:

On behalf of the Editor, I am pleased to inform you that your Stage 2 Registered Report RSOS-191288.R3 entitled "Modifying food items valuation and weight with gamified executive control training" has been deemed suitable for publication in Royal Society Open Science subject to minor revision in accordance with the referee suggestions. Please find the referees' comments at the end of this email.

The reviewers and Subject Editor have recommended publication, but also suggest some minor revisions to your manuscript. Therefore, I invite you to respond to the comments and revise your manuscript.

Please also ensure that all the below editorial sections are included where appropriate -- if any section is not applicable to your manuscript, please can we ask you to nevertheless include the heading, but explicitly state that the heading is inapplicable. An example of these sections is attached with this email.

- Ethics statement

If your study uses humans or animals please include details of the ethical approval received, including the name of the committee that granted approval. For human studies please also detail

whether informed consent was obtained. For field studies on animals please include details of all permissions, licences and/or approvals granted to carry out the fieldwork.

- Data accessibility

If you wish to submit your supporting data or code to Dryad (<http://datadryad.org/>), or modify your current submission to dryad, please use the following link:
[http://datadryad.org/submit?journalID=RSOS&manu=\(Document not available\)](http://datadryad.org/submit?journalID=RSOS&manu=(Document not available))

- Competing interests

- Authors' contributions

- Acknowledgements

- Funding statement

Because the schedule for publication is very tight, it is a condition of publication that you submit the revised version of your manuscript within 7 days (i.e. by the 07-May-2021). If you do not think you will be able to meet this date please let me know immediately.

To revise your manuscript, log into <https://mc.manuscriptcentral.com/rsos> and enter your Author Centre, where you will find your manuscript title listed under "Manuscripts with Decisions". Under "Actions," click on "Create a Revision." You will be unable to make your

revisions on the originally submitted version of the manuscript. Instead, revise your manuscript and upload a new version through your Author Centre.

on behalf of Professor Chris Chambers
(Registered Reports Editor, Royal Society Open Science)
openscience@royalsociety.org

Associate Editor Comments to Author (Professor Chris Chambers):

Associate Editor: 1

Comments to the Author:

One of the original Stage 1 reviewers was available to assess the Stage 2 manuscript, and I have decided to proceed on the basis of this review and my own reding. The assessment is positive while also some raising some methodological points requiring clarification, and some conceptual issues to consider in the Discussion. Please also attend to the note concerning the Abstract, which

I agree should provide a clear account of which hypotheses were tested and confirmed/ disconfirmed.

Provided the authors are able to respond thoroughly to these comments in a revision, full acceptance should be forthcoming without requiring further in-depth review.

Comments to Author:

Reviewer: 2

Comments to the Author(s)

Review RSOS

Dear Editor,

I only have a few comments, as this is a stage 2 registered report. I do not have the time to check their data analyses or perform a detailed comparison of documents. I trust the authors they did this adequately.

I think the most important question raised by the findings how effects can be found on value and not on weight. This raises a number of questions. First, are palatability ratings predictive for consumption? This seems logical but has this been tested before? Second, one explanation for why effects on weight are absent is that participants do not consume the trained products frequently. Can this be ruled out? This could parsimoniously explain why effects are found on value but not on weight. Another explanation could be that participants are not always responsible for the products and amount of food they consume as they may live together with others that are responsible for this. Can this explain the results?

I did not understand the description of the go/no-go task. The healthy foods are not mentioned in this description but are quite crucial. Moreover the difference between the experimental and control conditions is not explained. It is only understandable by looking at the tables.

Minor points:

Results for weight are not mentioned in the abstract. Moreover, it would be good to make explicit in the abstract how many hypothesis were tested and which were confirmed.

There are a number of paragraphs in the intro that consist of only one sentence.
Page 20 line 8; followed

Author's Response to Decision Letter for (RSOS-191288.R3)

See Appendix F.

Decision letter (RSOS-191288.R4)

Dear Dr Spierer:

It is a pleasure to accept your manuscript entitled "Modifying food items valuation and weight with gamified executive control training" in its current form for publication in Royal Society Open Science.

on behalf of Professor Chris Chambers (Subject Editor)
openscience@royalsociety.org

Appendix A

Review

MS title: Decreasing unhealthy food items valuation and weight with gamified executive control training

This is an interesting study proposal to explore the effect of gamified cognitive training on devaluation and weight loss. The study is in-keeping with recent literature in the field by presenting training via an app. The study also seeks to address the issue of expectancy effects and neatly proposes a design that is double blind throughout analysis.

I have one major concern with the proposed study, several comments that the authors may wish to consider and specific questions detailed below.

My main concern is with the question that the authors are trying to investigate. It is not clear from the report why the authors wish to combine GNG/ SST and CAT rather than investigating these training tasks in isolation? If results are positive, there is no way to determine why and if the results are non-significant it could be because the tasks interact with each other in ways we do not know about. I also have more specific concerns regarding the training tasks which are outlined below. The authors also emphasise the importance of controlling for expectancy effects, although this is not measured, and gamification, which is of questionable importance given recent literature.

The significance of the research question and the logic, rationale and plausibility of the hypotheses are therefore unclear.

The soundness / feasibility of the methodology and analysis are clear; however, see comments below.

The method section requires more detail to enable direct replication and prevent flexibility.

Positive controls should include analysis of learning and expectancy as detailed below.

Considerations

The introduction discusses behavioural stimulus interaction theory. This theory has recently been questioned by the authors (Chen et al., 2017, Appetite) and I recommend reviewing the following text as well:

McLaren, I. P. L., & Verbruggen, F. (2016). Association and inhibition. In R. A. Murphy, & R. C. Honey (Eds.), *The Wiley Blackwell handbook on the cognitive neuroscience of learning*. Chichester, England: John Wiley and Sons.

Expectancy effects are raised in the introduction but there is no mention of the studies that have analysed awareness and found no effects on outcomes. See Adams et al., 2016; Lawrence et al 2015 – refs 5 and 30.

Also see ref 30 who performed the same mediation analysis and found no effect.

The authors put a lot of emphasis on the gamification of the interventions but do not discuss the published research suggesting no effects of gamification on feasibility, acceptability or outcomes (refs 22 and 24).

The control group perform training with 50% mapping. It is possible that such training, which involves associative uncertainty, can lead to increased motivation and responding (Anselme et al., 2013, *Behavioural Brain Research*; Collins & Pearce, 1985, *JEP: APB*; Collins et al., 1983, *Quarterly Journal of Experimental Psychology*; Pearce & Hall, 1980, *Psychological Review*).

Specific questions

H2) states a smaller sample size if assumptions are not met – is this correct?

10 participants are added to account for dropout (~16%). I would expect this to be much higher given the study conditions (e.g. 21 minutes of training is double what we would normally have, 1m, 4m follow up) and strict analysis in/exclusion criteria (missing 2 days or not completing 5 days).

Page 9, final point in the inclusion criteria – please could you expand on this. Why do you require participants to like sweet and salty foods equally? Will you analyse the data immediately and exclude participants?

The most liked 50% of foods will be selected for each category – will you analyse whether the ratings for these foods are significantly different to those that are not selected? Or whether the 10% trim that you propose makes a difference?

If participants are coming into the lab pre- and post- intervention why not objectively record their weight. This is a more objective measure and could prevent loss of data from impossible values (participants in our studies have reported >7kg over a 2 week period).

The item valuation measure – does the pointer start at 0? Are there any major tickmarks or does the participant see their score while making the rating? If there is a maximum duration for responses why not programme this into the task?

The report would benefit from more figures e.g. for the procedure and tasks

Stimuli – you define un/healthy based on median scores from the foodpics database. Can the database be used in this way i.e. is there an even number of ‘healthy’ and ‘unhealthy’ foods. It would make more sense to have an absolute cut off based on caloric density. I also disagree that cheese is a neutral food item – I’m not aware of any cheeses that have a per 100g caloric density <198.

I also find it odd to include images of salt and sugar – I don’t expect many participants to crave pure salt/ sugar.

I think some of the foods coded as sweet/ savoury are ambiguous e.g. I would consider carrots / bell peppers to be fairly sweet.

There are 130 images detailed on the OSF with a lot of repeat images. It is not clear how this translates into the 100 images used in the study.

Training -

The training is designed to increase difficulty according to performance. However, the likely effect of such training is to maintain top-down control and prevent automatic inhibition. The current literature argues that automatic inhibition is likely to be more effective. See refs 5, 9, 14; Best et al., 2015, JEP: HPP.

It is assumed that the control training will control for expectancy effects – this needs to be measured and analysed or demonstrated through pilot testing.

GNG – are the instructions to drag items dependent on whether they are sweet / savoury i.e. half of the group would drag sweet and the other savoury? This is not clear.

Having to drag food items towards the bottom of the screen resembles an approach-avoid task. This response could either be considered an approach (self-reference AAT) or avoid (object-reference AAT).

It is unusual for a GNG task to have more go than no-go trials. This is inconsistent with previous training tasks and the idea that increasing successful control may be a key moderator for training effects (see ref 9)

Note that there may be issues for rewarding inhibition with feedback / points. See Guitart-Masip et al., 2012, Neuroimage.

SST – why is the SST so different from the GNG and CAT in terms of the ‘game’ with boxes and tracks?

Pg. 12 line 56 – the user responds on the incorrect side but they see green feedback and receive points – is this correct?

Table 2 would benefit from absolute numbers – I can’t work out the mapping based on un/healthy from this table.

Page 15, first paragraph. I don’t think I understood this, it might be useful to include a table with terms and definitions alongside some screenshots e.g. ‘total life increase’ ‘multiplier power’ This sounds like if the participant is performing well they can make the task easier which sounds counterintuitive

Analysis

Rather than excluding participants who miss training or have incomplete sessions why not look at dose-response instead?

Pg 18. Line 5 ‘healthiness rating’ – should this read ‘palatability rating’?

Pg 18. Line 19 – define ‘performance’

Pg 18. Line 30 – which 2 questionnaires? This is not clear from the methods

See my earlier comment on how self-reported weight can lead to impossible values – how will you deal with these?

One-sample t-tests should be included to see whether ratings significantly differ from 50 (i.e. there is a significant liking or disliking of foods). This may also be an important moderator for training effects

When looking at the difference between groups BMI makes more sense than weight – I suggest recording height as well to convert to baseline BMI.

Positive controls should include evidence of learning between training groups e.g. exp group should show increased % correct inhibitions compared to control group.

There also needs to be evidence that there is no difference in expectancy between groups

Timeline – how is it possible to analyse data during recruitment?

I believe that there are many reasons why this study could yield non-significant findings (multiple tasks, loss of data, matched expectancy effects(?)). I would suggest the authors use Bayesian analyses if they wish to make any inferences from such findings.

Appendix B

Response to reviewers, MS ID RSOS-191288

We thank the reviewers for their very constructive and helpful comments. Please find below our reply (in blue) and the main part modified in the manuscript (in green). All changes are also reported in blue in the new version of the manuscript.

REVIEWER #1

Dear editor,

Below is my review of the manuscript by Najberg and colleagues. Let me start by stating that the concept of registered reports makes great sense and I fully endorse this way of conducting research. I think the topic of the research is timely, and connects greatly to a recent recommendation to perform more well-powered preregistered experiments in the domain of ICT training (Carbine and Larson, 2019). Therefore, think the work lends itself really well for a registered report. I do have a number of comments that the authors may want to address before conducting the experiment. I have listed a number of issues and recommendations of how to improve the proposed research (at least in my opinion) below.

First, I think the decision to include the stop signal task requires more elaboration, and it may perhaps be wise to remove this training procedure from the intervention. The go/no-go task and cue-approach training have shown robust effects on food choice lasting for weeks to months (Chen et al., 2019; JPSP; Salomon et al., 2018; Scientific Reports), and food evaluation (e.g., Chen et al., 2016; JEP:GEN; Quandt et al., 2019; JEP:HPP) as suggested by well-powered preregistered experiments. However, as I understand it, it is less clear whether the stop signal training leads to behavior change, or changes in food evaluation, and there are some slight indications in the literature that this training might not work as well (Jones et al., 2016; Appetite; Veling et al., 2017; Curr. Add. Reports). That is because during stop signal training participants often erroneously respond to the stop items (due to the staircase procedure of presenting the stop signals late during a trial) which may counteract effects of the other training procedures (not responding to no-go items). Note that this is different from cue approach training and go/no-go training where participants never respond to the no-go (or uncued) items.

Together with the need to increase the diversity of the intervention to prevent boredom, we initially included the SST to propose an intervention involving all the main mechanisms thought to support changes in stimulus valuation/consumption based on motor control tasks. Even if current evidence indeed suggest that SST might be suboptimal in this regard, there are still findings supporting it (see also the work by Wessel PMID 25313953 and 26579025).

Yet, we agree with the reviewer (and reviewer 2) that the SST might not only have counterproductive effect, but its inclusion would also complicate mechanistic inferences on our intervention. We have thus followed the recommendation of the reviewers and removed the SST from our intervention.

Second, I think I do not quite understand the power analyses, as I was surprised by the modest proposed sample size. Recent work on cue-approach training and go/no-go training reporting power analyses suggest that in order to find effects on food choice in single session experiments employing a within-subjects design using repeated measurements of food choice between go (or cued) and no-go (or uncued) items 25 or more participants are required (Solomon et al., 2018; Chen et al., 2019). Moreover, effects of go/no-go training on food evaluation in a within design requires 30 participants to arrive at a power of .80 (e.g., Chen et al., 2016). Here the authors want to compare two between-subjects conditions, observe effects after some delay, and do not have any within-subjects baseline measures for the palatability ratings (as I understand it; see also my

comment on how this may be addressed). Based on these recent findings, I cannot imagine 25 participants per group will be sufficient to detect a decrease in evaluations between (rather than within) the groups.

Our Monte Carlo simulation approach enables determining power based on the specific variance parameter of the task of interest, as opposed to the more generic analytical approach used by e.g. G-power (Muthén & Muthén, 2002 and Paxton et al., 2001 in Structural Equation Modeling Journal). We didn't rely on previous literature to estimate relative effect sizes of interest since they are dependent on each studies parameter (e.g. within- and between- subject variance, number of trials etc.; e.g. Baker et al, 2019 arXiv:1902.06122), and may be over-inflated due to publication biases (see Button et al., 2013, Nat. Rev.: Neuro.; Nuijten et al., 2015, Rev. of Gen. Psy.). While the between-subject variance was reported in the literature for the rating and could be extracted from our previous studies for the weight, we did not have raw data to calculate the within-subject variances, and thus used the most reasonable estimate to run our simulations. Yet, we have now reconsidered our calculations based on the point raised by the reviewer and opted for a more conservative (higher) within-subject variance parameter to ensure reaching enough power (n= 86).

As I understand cue-approach training, it is a training to boost attention to go (or cued) items. The training does not aim to reduce responses to no-go (uncued) items. Empirically, all cue-approach training experiments that I have seen always compare go (cued) items with no-go (uncued items) leaving unanswered the question whether the training increases responses to go items or decreases responses to no-go items. In contrast, effects of go/no-go training on palatability ratings have been evaluated against a neutral (untrained) baseline, and effects appear to be driven by decreased evaluations of no-go items, except when a staircase on go trials is used in which evaluations of go items may go up (Chen et al., 2016; Experiment 2). Based on these findings and theoretical grounds, I would therefore predict that evaluations for go healthy foods would go up (as a result of cue approach training and staircase on go trials in go no-go task) and evaluations of no-go foods would go down (as a result of go/no-go training). Have the authors considered this possibility? The introduction may also benefit from discussing possible different working mechanisms of the different training procedures.

We thank the reviewer for raising this point. The Go/NoGo and the CAT may indeed influence responses to both Go and NoGo items, though in opposite direction and with putatively different amplitude. We mention this issue in the introduction and have adapted our design accordingly, as detailed in our replies to the points below.

It notably reads now, p. 3: "Of note, responses to Go stimuli during the Go/NoGo task and the withholding of responses to the non-cued stimuli during the CAT may also respectively develop approach (Chen et al., 2016; Experiment 2) or avoidance tendency (Schoenberg et al., 2014; Bakkour et al., 2016; Bakkour et al., 2017)."

Based on the arguments above I think it might be good to employ untrained baseline items in the design for both go healthy foods and no-go unhealthy foods. This may not only increase sensitivity to detect effects, but also aid with interpretation of the data on whether effects are more likely driven by increased ratings of healthy food items, or decreased ratings of unhealthy items. One way of doing this would be to apply matched (based on initial ratings) randomization of the food items into trained and untrained healthy food items and trained and untrained unhealthy foods. The statistical design would then be intervention (exp vs control) x session x item type (untrained trained) x food type (healthy vs unhealthy) where the last three factors are within-participants factors. It would require sufficient items per within-subject condition.

We were reluctant to include baseline items because:

- i) we could not exclude a generalization of the effect of training to untrained items of the same category. Such transfer would not be problematic in itself (it is even actually positive for real-life application), but may

complicate interpretations because the extent of a putative generalization is not clear. In the ‘worst’ case, a far transfer of the effects of training could take place and modify the valuation of all unhealthy items, potentially also including some baseline items.

- ii) We could not exclude different patterns of generalization for the unhealthy than healthy items, which could confound any difference between the effect on the Go and on the NoGo.
- iii) Adding the factor ‘item type’ would lead to target a four way interaction which would not only complicate the interpretation of an already complex design, but also increase the risk of false positive by multiplying the number of tests, reduce the statistical power of our primary contrasts of interest, and as pointed out by the reviewer, require finding enough ‘close but different’ healthy and unhealthy items.
- iv) Including additional items would double the duration of the rating phase, which could result in a decrease of the sensitivity of this measure (due to boredom, less engagement of the genuine valuation effort, etc).
- v) The baseline and trained items would need to be matched in terms of participants’ preference, which would be very difficult to set up given the limited number of possible (discriminable) stimulus category and of ‘preferred items’.

For these reasons, we preferred addressing the dissociation between the effect of training to NoGo and Go items using our control group. While our active control group will be matched with the experimental group in terms of item exposure, task engagement/practice and motor control training, they should not develop any bias/change in valuation to the Go nor to the NoGo items because the S-R mapping rule will not be biased toward the healthy/unhealthy items during the training.

To address the point raised by the reviewer on the importance of examining the effects of the training on the healthy in addition to the unhealthy items, we thus propose to add the factor ‘Food type’ to our initial design. By doing so, whether the training modified differently the Go and NoGo items could be assessed with an Intervention (exp vs control) x Session x Food type (healthy vs unhealthy) interaction.

With this modification of the design, we recalculated our sample size with the expectation to detect with 0.9 power an increase of 5 points for healthy items at minima and reversely for unhealthy items:

1. for the triple interaction, 23 per group (n tot = 46) are necessary (fig. 1).
2. For the planned double interactions (Intervention x Session on separately the healthy and unhealthy palatability ratings) 43 per group (n tot = 86) are necessary (fig. 2).

figure 1. EOI : Effect of Interest, i.e., how little is the effect we want to detect at minima for the palatability ratings. Power of our 2x2x2 mixed interaction for sample size per group of n=20 to 60 with the following parameters: sd-between = 15, sd-within = 7, EOI = 5, alpha = 0.05.

figure 2. EOI : Effect of Interest, i.e., how little is the effect we want to detect at minima for the palatability ratings. Power of our 2x2 mixed interaction for sample size per group of $n=20$ to 60 with the following parameters: $sd\text{-between} = 15$, $sd\text{-within} = 7$, $EOI = 5$, $\alpha = 0.05$.

Have the authors also considered the possibility of including a series of binary choices (e.g., as in Schonberg et al., 2014; Salomon et al., 2018; Chen et al., 2019) between go and no-go items pre and post? Work suggests both cue-approach training and go/no-go training show robust effects on such measurements (see citations above), and there is even one experiment showing that the probability of choosing healthy over unhealthy can be increased by training (Chen et al., 2019). Here it would also be possible to include baseline (untrained) choice pairs.

Since participants already have to rate 100 items before the intervention, we did not include further assessments of palatability because we considered it would result in too much repetition. We felt that the participants would eventually not make the effort to correctly perform the task, which would decrease the sensitivity of our primary outcome measure. Yet, should the reviewer think that the benefits of binary choices would outweigh the costs of including additional rating time, we would agree to implement this second type of assessment at the pre- and post-intervention sessions.

It would be better if it is possible to measure weight objectively, but perhaps that is not possible. A choice measure as described above might be another good measure of behavior.

Participants will actually measure their weight objectively since they will be instructed to use a scale at home or in a pharmacy. We also included strict procedural instructions to improve the reliability of the measure: the participants will have to weigh themselves just after waking up, before eating and after going to the toilet, and to use the same scale for all measures.

It would also complicate our recruiting if the participants have to come at the first hour of the morning to our laboratory at pre- and post-intervention, as well as 1 and 4 months after the end of the intervention.

Gamifying training tasks might be risky because subtle changes may influence effects. For instance, by changing some of the task parameters (e.g., employing a staircase in go / no-go training) attention may shift toward certain item types (e.g., more attention to go items than no-go items) which may influence palatability ratings

in some way (e.g., leading to stronger go valuation instead of no-go devaluation; see e.g., Quandt et al., 2019; JEP:HPP). In light of this I wondered whether the authors have considered to pilot the gamified intervention in a single session experiment to be at least sure the task works on the short-term to influence palatability ratings (my experience is that go/no-go virtually always works to decrease evaluations of no-go versus go items from pre to post when the evaluations are matched on the pre-rating). Perhaps it is possible to test this first before offering these tasks for a more intensive training especially in light of the fact that the new gamified tasks are slightly modified from the original tasks.

We agree with the reviewer that piloting the intervention would help limiting the risk of null results. Yet, for the reasons listed below, we thought that the cost/benefit ratio of such approach would be too high:

- i) Drawing inference on the intervention effectiveness based on the pilot study would require it to be well powered (underpowered study could yield both over- or under- inflated effect size, e.g. PMID: 23571845 or 18633328).
- ii) It is not clear whether the effects of short single-session interventions are comparable to those of longer multi-sessions interventions, and thus if a result of a short pilot intervention would be generalizable to our planned long-term intervention.
- iii) ICT has been proven effective with a variety of tasks and context by different groups. We consider the modifications we introduced to be in the range of this variability.
- iv) The key 'ingredient' of the ICT training (i.e. inhibiting/executing motor responses to the target items) remained unchanged even if the training environment is gamified. The gamification mostly consisted in increasing participants immersion, motivation and reducing boredom. Hence, the gamification would not prevent concluding on whether motor control influences stimuli valuations/weight because this mechanism remained the core of the training task. In this regard, we would further note that if this mechanism is so sensitive to the training context that it does not take place anymore when motivational elements are added, it is probably not existing nor worth investigating.
- v) Partly ensuring the adequacy of our intervention, the present intervention has been used in a 3-week intervention trial to improve inhibitory control in healthy young vs older population (Najberg et al, in preparation), and while the aim was not to change items' perceived value, it allowed ensuring that our gamified tasks indeed induce RT and FA rate in a typical range for such tasks.

I am not sure about removing palatability ratings slower than 1500 (this may be needed for choices but not ratings; for a discussion see Chen et al., 2019).

We thank the reviewer for his suggestion and removed this exclusion rule.

Maybe I missed it, but do the authors also measure possible differences in expectations between the intervention and control group? This seems important because this is one of the major limitations they seem to address.

We apologize for our lack of clarity and details on this question. We actually already planned to measure expectations between the experimental and control groups, but referred to it as the 'blinding variable' (assessed with the post-intervention debriefing questionnaire). We have now clarified this point and also propose a better assessment of expectancy following Boot and colleagues (2013: PMID 26173122) recommendations of using two binary variables on both weight and palatability rating measures in addition to odds ratios analyses. We further included these variables as positive controls in the design.

It reads now p. 15: “In this questionnaire, participants' expectations on the effects of the experimental and control interventions are assessed with the following question: “What do you think the game practice has improved or modified?”. If the response includes the notion of better eating habit and/or modification of items ratings (healthy items up and unhealthy items down), the variable "Expected valuation" will take the value 1. If the response includes notion of weight loss, the variable "Expected weight loss" will take the value 1. Any other responses (not understanding the outcome, expectation of weight gain, unhealthy item valuation increase, healthy item valuation decrease, etc.) will result in the value 0 in their respective variables.”, and p. 19: “Expectancy effect will be measured based on the count data of both "Expected valuation" and "Expected weight loss" binary variables (see Questionnaires section) on two 2-by-2 contingency matrices, with the phi as effect size. In case of a $\phi < 0.2$, we will replace randomly chosen "unblinded" and "blinded" participants from the most divergent count-value until we reach a $\phi > 0.2$ or equal to 0.2. For example, in the case we have 8 "blinded" participants in the control group (35 "unblinded") and 1 "blinded" participant in the experimental group (42 "unblinded"), we will replace randomly one of the 8 in the control group by a new participant.

The result of the chi-square test (or fisher exact test if a cell have a count data below 5) will be reported to better understand the feasibility of this method to create matching expectations. However, we still support brut forcing the matching expectancy to avoid our key contrasts to be confounded by this variable.”

I hope my comments and suggestions are useful to further improve the proposed research. I look forward to learn about the final experimental protocol and results.

Signed review,

REVIEW #2

MS title: Decreasing unhealthy food items valuation and weight with gamified executive control training This is an interesting study proposal to explore the effect of gamified cognitive training on devaluation and weight loss. The study is in-keeping with recent literature in the field by presenting training via an app. The study also seeks to address the issue of expectancy effects and neatly proposes a design that is double blind throughout analysis. I have one major concern with the proposed study, several comments that the authors may wish to consider and specific questions detailed below.

My main concern is with the question that the authors are trying to investigate. It is not clear from the report why the authors wish to combine GNG/ SST and CAT rather than investigating these training tasks in isolation? If results are positive, there is no way to determine why and if the results are nonsignificant it could be because the tasks interact with each other in ways we do not know about.

Identifying whether and how motor control influences stimuli valuation and in turn consumption would indeed require highly simplified and isolated tasks to delineate the condition and mechanisms for training-induced changes in valuation and consumption. Yet, these questions have already been the focus of more than 20 studies (see e.g. PMID: 31122177 and 26592707 for review on the effect of inhibitory control training (ICT) on food). This literature indicates that ICT have robust and reproducible effects, even if the training were based on variable types of training tasks and contexts. Of note, while still scarce, previous findings from training combining several tasks seem very promising (e.g. Stice et al., 2017).

On this basis, the aim of our study is first to confirm that motor control influences valuation with long-term intervention (most of the existing literature used single training session of 30 to 60 minutes) and that this effect may account for putatively associated weight loss. Critically, we aim to do so with a stringent control for expectations (control group differing only in terms of the active SR mapping rule 'ingredient') and with a motivating practice environment. This design would allow to respectively demonstrate the hypothesized mechanisms are indeed SR mapping rules (and not merely improving top-down control by practicing inhibitory control task) and moving one step further the development of intervention that could be proposed in population willing to modify healthy/unhealthy food items valuation and consumption balance (i.e. gamified, online interventions).

Hence, we feel that assessing with double-blinded pre-registered design an intervention combining well-established training approaches and presenting it in an online gamified version still represents an important step forward in this field of research.

We however agree that our interpretation may benefit from the intervention being simplified and the study design remodeled. For this reason, and also in line with the comments of Rev. 1, we removed the SST from the intervention and modified our design to contrast the effects of the intervention on both the Go and on the NoGo, helping to better delineate the effects of the intervention (via an Intervention (exp vs control) x Session (pre-; post- training) x Food type (healthy vs unhealthy) interaction).

Our study will mainly allow concluding i) if there is a causal link between the repetition of motor control to specific items and a change in their valuation, ii) if this effect differs when participants inhibit or execute motor responses to the items, and iii) whether expectations confounded previous investigation of these questions.

I also have more specific concerns regarding the training tasks which are outlined below.

The authors also emphasise the importance of controlling for expectancy effects, although this is not measured, and gamification, which is of questionable importance given recent literature.

We agree and we apologize for our lack of clarity and details on the question of expectancies. We actually already planned to measure expectations between the experimental and control groups, but referred to it as the 'blinding variable' assessed with the post-intervention debriefing questionnaire. We have now clarified this point and also propose a better assessment of expectancy following Boot and colleagues (2013: PMID 26173122) recommendations of using two binary variables on both weight and palatability rating measures in addition to odds ratios analyses. We further included these variables as positive controls in the design.

Regarding the gamification, by collaborating with professional videogame developers and grounding our gamification choices on models from the literature (e.g. Schell J., 2008, *The Art of Game Design: A Book of Lenses*. San Francisco, CA, USA., and Hunicke R, LeBlanc M, et al., *MDA: A formal approach to game design and game research*, aaii.org), we have reached an unprecedented level of gamification for ICT interventions (including arts, local and global rankings, in-game economy, and power-ups), which make comparisons with the effects of previous attempts to gamify ICTs difficult. Critically, the implemented gamification mechanisms do not modify the task mechanics; the 'active ingredients' of ICT and CAT (the biased control of the motor response to specific items) remained unchanged. The gamification 'only' aims at improving participants immersion, motivation and long-term adherence, three important aspects to foster the targeted plastic changes and the long-term adoption of the training program in healthy and clinical populations.

The soundness / feasibility of the methodology and analysis are clear; however, see comments below. The method section requires more detail to enable direct replication and prevent flexibility. Positive controls should include analysis of learning and expectancy as detailed below.

We thank the reviewer for his/her positive and constructive evaluation and address his/her points in details below.

Considerations

The introduction discusses behavioural stimulus interaction theory. This theory has recently been questioned by the authors (Chen et al., 2017, *Appetite*) and I recommend reviewing the following text as well: McLaren, I. P. L., & Verbruggen, F. (2016). Association and inhibition. In R. A. Murphy, & R. C. Honey (Eds.), *The Wiley Blackwell handbook on the cognitive neuroscience of learning*. Chichester, England: John Wiley and Sons.

Indeed, we thank the reviewer for having pointed out this relevant debate and literature. We have now developed the section on the associative learning mechanisms advanced in the text suggested by the reviewer. In addition, since the SST task has been removed and that we now more specifically focus on the Go/NoGo task we have in line with the suggestion of the reviewer removed the mention of the BSI in the Introduction.

It notably reads now, p. 3: "In Go/NoGo tasks, participants are instructed to respond as fast as possible to a given category of items and to withhold their responses to another category. Such practice actually acts as motivational conditioning paradigm which eventually automatizes the engagement of inhibition processes via associative learning mechanisms and reduces its perceived value. When a NoGo stimuli becomes associated with avoidance/aversion, its presentation would directly trigger the avoidance/aversive center, which would then suppress the activation of the approach/appetitive center (Dickinson and Balleine, 2002). This mechanism is thought to eventually result in the decrease of the hedonic and motivational value of the NoGo stimuli (Ferrey et al., 2012; Houben et al., 2012; Veling et al., 2008; McLaren and Verbruggen, 2016)."

Expectancy effects are raised in the introduction but there is no mention of the studies that have analysed awareness and found no effects on outcomes. See Adams et al., 2016; Lawrence et al 2015 – refs 5 and 30.

We thank the reviewer for his/her suggestion. We now discuss this literature in the Introduction section. It reads now, p. 5: “Previous evidence for an absence of effect of participants’ awareness of training goals on food-ICTs effectiveness (Adams et al., 2016; Lawrence et al 2015b) indicate that this approach should not influence the efficacy of our experimental intervention”

Also see ref 30 who performed the same mediation analysis and found no effect.

We now quote the reference suggested by the reviewer in our section on the mediation analysis (p. 18). Yet, we do not feel that this sole negative finding justifies abandoning this approach. Including a mediation analyses in the present study would not only have a replicative value, but since our study is strongly powered and involves a slightly different approach, we feel that it may still help identifying whether or not decreases of palatability with ICT is associated with weight reduction.

The authors put a lot of emphasis on the gamification of the interventions but do not discuss the published research suggesting no effects of gamification on feasibility, acceptability or outcomes (refs 22 and 24).

We agree that this aspect deserves more discussion. As suggested by the reviewer, we have now further developed this section in the introduction.

It reads now, p. 4: “While some research suggest that improving the practice environment with task gamification did not improve training feasibility, acceptability or outcome (Forman et al., 2019; Poppelaars et al., 2018), the effect of most of previous studies may still have been minimized by presenting task in weakly motivating practice environments (Forman et al., 2019; Poppelaars et al., 2018).”

The control group perform training with 50% mapping. It is possible that such training, which involves associative uncertainty, can lead to increased motivation and responding (Anselme et al., 2013, Behavioural Brain Research; Collins & Pearce, 1985, JEP: APB; Collins et al., 1983, Quarterly Journal of Experimental Psychology; Pearce & Hall, 1980, Psychological Review).

We thank the reviewer for raising this issue. We now mention this possibility in the Introduction section. Of note, we also induce such associative uncertainty in the experimental group with the use of “neutral” items that appears both in the Go and NoGo trials. In spite of the possibility for such effects to emerge differently between the experimental and control group, we feel that using unbiased S-R mapping rules for the control group was the optimal way to modulate the key ‘active ingredient’ of the intervention while controlling for expectations.

It reads now, p. 5: “Furthermore, to minimize the potential effect of the larger associative uncertainty in the control than experimental group on motivation and responding (Anselme et al., 2013; Collins & Pearce, 1985; Collins et al., 1983; Pearce & Hall, 1980), and participants' expectations on the aim of the study, neutral distractors food items will be added to both the experimental and control group.”

Specific questions

H2) states a smaller sample size if assumptions are not met – is this correct? 10 participants are added to account for dropout (~16%). I would expect this to be much higher given the study conditions (e.g. 21 minutes

of training is double what we would normally have, 1m, 4m follow up) and strict analysis in/exclusion criteria (missing 2 days or not completing 5 days).

We agree that the number of drop out is difficult to estimate; we have thus modified our approach for dealing with dropouts by simply replacing all excluded participants by new ones. Since an external lab manager not further involved in the study will be in charge of the condition assignments, we could replace participants one by one while leaving the sample size balanced between the experimental and control groups.

Page 9, final point in the inclusion criteria – please could you expand on this. Why do you require participants to like sweet and salty foods equally? Will you analyse the data immediately and exclude participants?

To minimize participants awareness of the aim of the study, we wanted the task instructions (i.e. focus on sweet/salty) to be orthogonal to the effect of interest (modifying healthy/unhealthy valuation). Hence, to prevent any confounds related to the intervention-irrelevant sweet/salty dimension, we had to balance baseline palatability rating between the sweet and salty food. We planned to measure the rating of each group of stimuli using a short questionnaire (link to the OSF) and to analyze the results directly during the screening session as it only involves pasting 10 numbers in an excel macro.

We have now clarified this point p. 7: “Since participants will have to discriminate the Go and NoGo trials using a sweet/salty rule in the Go/NoGo task, we balanced the liking of both types of items.”

The most liked 50% of foods will be selected for each category – will you analyse whether the ratings for these foods are significantly different to those that are not selected? Or whether the 10% trim that you propose makes a difference?

Since with these selection criteria we artificially split a continuous distribution, we felt it would be statistically incorrect (tautologic) to then test for the hypothesis of a difference (see e.g. Albeson’s Statistics as Principled Argument, 1995). As for the trimming, since this method does not change the position of bulk of the distribution it should not induce any difference.

If participants are coming into the lab pre- and post- intervention why not objectively record their weight. This is a more objective measure and could prevent loss of data from impossible values (participants in our studies have reported >7kg over a 2 week period).

Participants will actually measure their weight objectively since they will be instructed to use a scale at home or in a pharmacy. We also included strict procedural instructions to improve the reliability of the measure: the participants will have to weigh themselves just after waking up, before eating and after going to the toilet, and to use the same scale for all measures.

It would also complicate our recruiting if the participants have to come at the first hour of the morning to our laboratory at pre- and post-intervention, as well as 1 and 4 months after the end of the intervention.

The item valuation measure – does the pointer start at 0? Are there any major tickmarks or does the participant see their score while making the rating? If there is a maximum duration for responses why not programme this into the task? The report would benefit from more figures e.g. for the procedure and tasks

We included a marker in the middle and minimum-maximum anchors of our analogue scales. A blue arrow indicates where the participants touch the screen. According to the reviewer's suggestion, we have now removed the maximum response window limit and now provide figures to facilitate the evaluation of the study tasks.

Stimuli – you define un/healthy based on median scores from the foodpics database. Can the database be used in this way i.e. is there an even number of 'healthy' and 'unhealthy' foods. It would make more sense to have an absolute cut off based on caloric density.

The Blechert et al. database contains food pictures from all categories in equal amounts, and our target kcal density variable has a low skewness (0.54). There was thus indeed the same number of 'healthy' and 'unhealthy' items and the median could be interpreted. Accordingly, we felt more appropriate to determine cutoffs for healthy and unhealthy items based on the database quartiles (≤ 49.9 kcal and 198 kcal).

I also disagree that cheese is a neutral food item – I'm not aware of any cheeses that have a per 100g caloric density <198 .

The cheese is indeed above our calorie density cut off for unhealthy items. Yet, we did not consider in isolation the cutoffs for unhealthy and healthy items to select the neutral items because in this case it was critical to also take into account the *quantity* of the items typically ingested in our target population (see Stimuli paragraph of the method section).

Accordingly, we considered the cheese items as neutral because in Switzerland, non-melted cheese is typically not overconsumed and thus not considered as an unhealthy item.

I also find it odd to include images of salt and sugar – I don't expect many participants to crave pure salt/ sugar.

Salt and sugar were included as neutral items fillers/distractors to improve the matching between the experimental and the control conditions (and no palatability ratings will be measured on these items). Since the pictures of pure salt and sugar are in this neutral items category, we aimed at having no craving associated with these items.

I think some of the foods coded as sweet/ savoury are ambiguous e.g. I would consider carrots / bell peppers to be fairly sweet.

We agree, but we don't think it is problematic in our design. The sweet / salty categorization task is indeed orthogonal to the aim of the study and was used to distract the participants from the intervention-relevant items healthiness while ensuring that they actually processed/identified the items during the tasks. Accordingly, some errors/learning in the salty/sugar categorization is in our view tolerable. That said, if the reviewer considers that an obvious salty/sugar categorization for each item is necessary, we would of course agree to remove the most ambiguous items from the list.

There are 130 images detailed on the OSF with a lot of repeat images. It is not clear how this translates into the 100 images used in the study.

We apologize for our lack of clarity. There are 50 unhealthy items, 50 healthy items and 30 neutral items. The 30 neutral items will not be rated by the participants, they are only distractors. This is now clarified in the manuscript.

Training - The training is designed to increase difficulty according to performance. However, the likely effect of such training is to maintain top-down control and prevent automatic inhibition. The current literature argues that automatic inhibition is likely to be more effective. See refs 5, 9, 14; Best et al., 2015, JEP: HPP.

The difficulty progression curve is designed so that the majority of the trials is easy (i.e. before reaching challenging difficulty levels, the participants will have to go through an important phase with loose difficulty parameters). In addition, since the total training time will be of ca 20 hours and the number of different items limited, there will be a lot of repetitions of each stimulus-response association (tot. > 3500 trials for the GNG and > 3300 trials for the CAT). We thus think that even if some phases of the task will be challenging, automatic inhibition will develop.

It is assumed that the control training will control for expectancy effects – this needs to be measured and analysed or demonstrated through pilot testing.

We agree and we apologize for our lack of clarity and details on the question of expectancies. We actually already planned to measure expectations between the experimental and control groups, but referred to it as the 'blinding variable' assessed with the post-intervention debriefing questionnaire. We have now clarified this point and also propose a better assessment of expectancy following Boot and colleagues (2013: PMID 26173122) recommendations of using two binary variables on both weight and palatability rating measures in addition to odds ratios analyses. We further included these variables as positive controls in the design.

It reads now p. 15: "In this questionnaire, participants' expectations on the effects of the experimental and control interventions are assessed with the following question: "What do you think the game practice has improved or modified?". If the response includes the notion of better eating habit and/or modification of items ratings (healthy items up and unhealthy items down), the variable "Expected valuation" will take the value 1. If the response includes notion of weight loss, the variable "Expected weight loss" will take the value 1. Any other responses (not understanding the outcome, expectation of weight gain, unhealthy item valuation increase, healthy item valuation decrease, etc.) will result in the value 0 in their respective variables.", and p. 19: "Expectancy effect will be measured based on the count data of both "Expected valuation" and "Expected weight loss" binary variables (see Questionnaires section) on two 2-by-2 contingency matrices, with the phi as effect size. In case of a $\phi < 0.2$, we will replace randomly chosen "unblinded" and "blinded" participants from the most divergent count-value until we reach a $\phi > 0.2$ or equal to 0.2. For example, in the case we have 8 "blinded" participants in the control group (35 "unblinded") and 1 "blinded" participant in the experimental group (42 "unblinded"), we will replace randomly one of the 8 in the control group by a new participant.

The result of the chi-square test (or fisher exact test if a cell have a count data below 5) will be reported to better understand the feasibility of this method to create matching expectations. However, we still support brut forcing the matching expectancy to avoid our key contrasts to be confounded by this variable."

GNG – are the instructions to drag items dependent on whether they are sweet / savoury i.e. half of the group would drag sweet and the other savoury? This is not clear.

The Go and NoGo stimuli will be either sweet or salty items: for the two groups. Each time a participant starts the GNG task, the Go trials has a 0.5 chance to be salty and a 0.5 chance to be sweet.

This issue has now been clarified in the method section. It reads now, p.11: “In the Go/NoGo task, participants will be presented with food pictures and instructed to drag as fast as possible toward the bottom of the screen either the sweet items (e.g. orange, ice-cream; half of the training sessions) or the salty items (e.g., green beans, hamburger; other half of the sessions).”.

having to drag food items towards the bottom of the screen resembles an approach-avoid task. This response could either be considered an approach (self-reference AAT) or avoid (object-reference AAT).

We introduced the dragging movement to slow down the task a little. Yet, participants have to drag the item on a distance of min 1cm, which we consider would unlikely be interpreted by the participants as an approach behavior.

It is unusual for a GNG task to have more go than no-go trials. This is inconsistent with previous training tasks and the idea that increasing successful control may be a key moderator for training effects (see ref 9). Note that there may be issues for rewarding inhibition with feedback / points. See Guitart-Masip et al., 2012, Neuroimage.

We prefer to use the typical Go/NoGo task ratio (more Go than NoGo trial) and not the usual Go/NoGo *training* tasks ratio (equiprobable trial types) to ensure a strong response prepotency, which is together with response time deadline the second parameter allowing to create a demand for response inhibition during NoGo trials (if there is no response tendency, there is nothing to inhibit during NoGo trials and then the task become a response selection task). Since the inhibition of motor responses is considered as the key mechanisms to induce stimulus devaluation, we aimed at maximizing the involvement of this process in our task by presenting more Go than NoGo and including a response time threshold.

While we acknowledge that 50/50 Go/NoGo would have increased the number of successful inhibition trials and the occurrence of the NoGo stim-inhibition association, we felt that the chosen NoGo ratio was the most appropriate choice because it improves the demand for inhibition while still allowing participants to successfully inhibit responses to NoGo ca 85% of the time (a ratio corresponding to the typical 15% false alarm rate targeted in Go/NoGo tasks).

This choice is also in line with findings that the devaluation effect induced by the GNG task is observed when the no-go trials are rare, but may decrease when they are frequent (Chen et al., 2016: JEPG).

As for the feedback, it was necessary to provide participants with information on their performance to ensure they had understood the instructions and to motivate them reaching higher scores. The feedback also represents an important part of the gamification because it is linked with the scores. We however now discuss the potential effects of rewarding inhibition with feedback and quote the literature kindly pointed out by the reviewer.

It reads now, p. 11: “While feedback on performance may interact with the effect of training (Guitart-Masip et al., 2012), it allows ensuring a correct understanding of the instruction by the participants and using scores as gamification parameters.”.

SST – why is the SST so different from the GNG and CAT in terms of the ‘game’ with boxes and tracks?

Pg. 12 line 56 – the user responds on the incorrect side but they see green feedback and receive points – is this correct?

According to the comments of the reviewer (see also above), we have now removed the SST from the intervention.

Table 2 would benefit from absolute numbers – I can't work out the mapping based on un/healthy from this table.

Since the total number of items will vary based on the participants' performance/playing time, including the exact number of trials for each participant was not possible and we thus used percentage. Yet, to clarify this aspect in the manuscript, we have now improved the table 2 organization to facilitate the mapping.

Page 15, first paragraph. I don't think I understood this, it might be useful to include a table with terms and definitions alongside some screenshots e.g. 'total life increase' 'multiplier power' This sounds like if the participant is performing well they can make the task easier, which sounds counterintuitive.

With the amount of time participants will play, they will be able to progress further in the difficulty curve. However, this does not make the task easier in itself. They are just allowed slightly more errors before the end of the training sessions. We have now uploaded as supplementary material more details about the intervention gamified mechanics on the study OSF page.

Analysis

Rather than excluding participants who miss training or have incomplete sessions why not look at dose-response instead?

We expect very few dropouts from missing training and we will be able to detect it immediately since the data will be automatically uploaded daily on our server, which we will monitor. In this case, since dropouts are typically the sign of a disengagement, we could not infer they really did the effort to complete the task as expected during the recorded part of the training, and thus whether they could be compared to other participants.

Pg 18. Line 5 'healthiness rating' – should this read 'palatability rating'?

This error has been corrected.

Pg 18. Line 19 – define 'performance'

This is now clarified. It now reads, p.16: "This performance will be defined by the participants' average score per day without the effect of the power-ups. This value best represents the performance as it does the interaction between the accuracy and the reaction time induced by the difficulty rate".

Pg 18. Line 30 – which 2 questionnaires? This is not clear from the methods

In this section, we are referring to the pre- and post-intervention palatability rating questionnaires. It reads now p.16: "if the pre- or the post-intervention palatability rating questionnaires are not completed".

See my earlier comment on how self-reported weight can lead to impossible values – how will you deal with these?

As detailed in the data exclusion section of the method, outlier participants will be defined as those outside the $2.5 * MAD$ (median average deviation; moderately conservative criterion) range around the median (Leys et al., 2013: JESP). Additionally, we now follow the reviewer comment and take into account the distributions of the between-measurements deltas.

It now reads p. 17: “Changes in weight will be assessed using three deltas: pre- vs. post-Intervention; post-Intervention vs. 1-Month follow-up; 1-Month vs. 4-Month follow-up. For each distribution of these deltas, outlier changes in weight values will be defined as those outside the $3 * MAD$ range around the median, and the concerned data points will be excluded from the analyses.”

One-sample t-tests should be included to see whether ratings significantly differ from 50 (i.e. there is a significant liking or disliking of foods). This may also be an important moderator for training effects

We thank the reviewer for this comment, we have now included this suggestion in our positive controls. However, we prefer to only report an effect size (as for our other positive controls), since a t-test would only allow identifying if the effect can be generalized to the general population, but not better capturing the characteristic of the sample than an effect size (e.g. Lakens D., 2013: Front. Psychol.).

It now reads p. 19: “A difference of Cohen’s $d < 0.2$ will be expected between the mean palatability ratings of the trimmed trained items and the middle of the analogue scale (i.e., 50 points). To ensure that the healthy and unhealthy items were actually liked by the participants (and not neutral), if a $d \geq 0.2$ difference is observed, participants with the value the farther from the pooled groups median will be successively excluded until the groups are balanced.”

When looking at the difference between groups BMI makes more sense than weight – I suggest recording height as well to convert to baseline BMI.

Agreed, we thank the reviewer for his/her advice, and we are now focusing on/reporting the BMI for the baseline measurement.

Positive controls should include evidence of learning between training groups e.g. exp group should show increased % correct inhibitions compared to control group.

We agree and now include the test suggested by the reviewer as a positive control. It reads now, p. 19: “Since both the experimental and control training should induce a corresponding amount of learning, a small difference of Cohen’s $d < 0.4$ will be expected for the delta pre- vs. post-Intervention performance (i.e., the score without the impact of the power-ups) between the two groups. If a $d \geq 0.4$ difference is observed, participants with the value the farther from the pooled groups median will be successively excluded until the groups are balanced.”

There also needs to be evidence that there is no difference in expectancy between groups

We agree and now include the test suggested by the reviewer as a positive control. It reads now, p. 19: "Expectancy effect will be measured based on the count data of both "Expected valuation" and "Expected weight loss" binary variables (see Questionnaires section) on two 2-by-2 contingency matrices, with the phi as effect size. In case of a $\phi < 0.2$, we will replace randomly chosen "unblinded" and "blinded" participants from the most divergent count-value until we reach a $\phi > 0.2$ or equal to 0.2. For example, in the case we have 8 "blinded" participants in the control group (35 "unblinded") and 1 "blinded" participant in the experimental group (42 "unblinded"), we will replace randomly one of the 8 in the control group by a new participant.

The result of the chi-square test (or fisher exact test if a cell have a count data below 5) will be reported to better understand the feasibility of this method to create matching expectations. However, we still support brut forcing the matching expectancy to avoid our key contrasts to be confounded by this variable."

Timeline – how is it possible to analyse data during recruitment?

Analyses during recruitment include all the preprocessing steps: fetching the data sent by the training application to our servers, transforming the android logs into readable datasets, and monitoring the datasets to verify data exclusion rules such as the amount of time played per training sessions.

I believe that there are many reasons why this study could yield non-significant findings (multiple tasks, loss of data, matched expectancy effects(?)). I would suggest the authors use Bayesian analyses if they wish to make any inferences from such findings.

We have now planned Bayes Factors in case of p-values below our alpha (0.05) to help the interpretation for null findings. It now reads p. 16: "In case of p-values above our 0.05 alpha threshold, Bayes Factors will be computed using the R BayesFactor package [58] to estimate the likelihood of the null hypothesis."

Appendix C

Review

MS title: Decreasing unhealthy food items valuation and weight with gamified executive control training

Thank you for addressing my comments/ concerns. I just have some minor comments regarding the revised manuscript.

My first review raised the question of why the authors want to combine GNG / CAT. My question was answered in the response but I still feel that some of this information is lacking in the intro. Currently the main aims are to address issues with task parameters and expectancies – but the combination of tasks seems to have been overlooked.

Pg 4, line 40 the authors discuss how GNG is expected to influence unhealthy eating and CAT to influence healthy eating but there is nothing in the intro at a broader level regarding the importance of combining tasks and the effects that this may have?

One of my questions in the initial review was with regards to the distinction between sweet / salty items, with some items being ambiguous (e.g. bell peppers). If the authors are looking at the training data itself (reaction times and % correct) as an indication of learning – this ambiguity could cause an issue unless there are an equal number of ambiguous healthy and unhealthy foods. I would guess that responses would be slower and there will be more errors in the healthy foods purely because some of the foods are more ambiguous.

My original review mentions one sample t-test to see whether ratings are significantly different to 50 on the VAS. In the response the authors quote a d of 0.2 but this is 0.3 in the manuscript. Please clarify.

In relation to this point I'm not sure I understand the logic here:

To ensure that the healthy and unhealthy items were actually liked by the participants (and not neutral), if a $d \geq 0.2$ difference is observed, participants with the value the farther from the pooled groups median will be successively excluded until the groups are balanced.

Wouldn't a greater d indicate that responses were further from 50 and therefore liked/ disliked and not neutral? If you exclude participants the greater scores you will be excluding the participants who liked/ disliked the foods the most?

I also don't follow the logic that a similar amount of learning is expected in the control and active groups considering the difference in mapping. If the training does not induce differences in learning then how can this be used as a manipulation check?

“Since both the experimental and control training should induce a corresponding amount of learning, I'm again not sure why you would remove participants who show the greatest learning effects – this is potentially very informative and could be an important mediator?

Manuscript

Could you please clarify your hypotheses:

H1b): the reverse pattern for the healthy items palatability ratings;

This is slightly ambiguous and could mean a smaller decrease for the exp group or a greater decrease for the control group.

- H3) the shift in items' valuation mediates the change in weight.

Is this for both unhealthy and healthy shifts i.e. 2 analyses to explore these changes separately? If so please state the 2 hypotheses in full.

For the power analysis, can you please provide more detail on the OSF in the form of a readme file. From the current files I couldn't determine whether the data from the 'previous datasets' mentioned in the comment below was available or what the data was (is this your own data or sourced from elsewhere, please provide references if the latter).

2. For each dependent variable separately, we identified the task-related between- and within- subject variance based on previous datasets from studies with the same approaches.

Will the custom-made general health questionnaire be made available, only there is very little information on this?

Appendix D

Response to reviewers, MS ID RSOS-191288.R1

We thank the reviewer for his/her latest comments. Please find below our reply (in blue) and the main part modified in the manuscript (in green). All changes are also reported in blue in the new version of the manuscript.

Review

MS title: Decreasing unhealthy food items valuation and weight with gamified executive control training

Thank you for addressing my comments/ concerns. I just have some minor comments regarding the revised manuscript.

My first review raised the question of why the authors want to combine GNG / CAT. My question was answered in the response but I still feel that some of this information is lacking in the intro. Currently the main aims are to address issues with task parameters and expectancies – but the combination of tasks seems to have been overlooked.

Pg 4, line 40 the authors discuss how GNG is expected to influence unhealthy eating and CAT to influence healthy eating but there is nothing in the intro at a broader level regarding the importance of combining tasks and the effects that this may have?

We agree. We have now included the following section in the introduction: It reads now, p. 3-4:

“On this basis, we reasoned that the development of automatic inhibition to unhealthy items and of attentional biases towards healthy items would act synergistically to eventually improve eating habits. This approach would indeed promote the replacement of unhealthy items by healthy ones and not merely reducing unhealthy item consumption, thereby ensuring to maintain satiety. This hypothesis is supported by previous findings for large decreases of reward responses to unhealthy items and body fat with multifaceted training approach involving both attentional bias modification and response inhibition approaches (e.g. Stice et al., 2017).”

One of my questions in the initial review was with regards to the distinction between sweet / salty items, with some items being ambiguous (e.g. bell peppers). If the authors are looking at the training data itself (reaction times and % correct) as an indication of learning – this ambiguity could cause an issue unless there are an equal number of ambiguous healthy and unhealthy foods. I would guess that responses would be slower and there will be more errors in the healthy foods purely because some of the foods are more ambiguous.

We agree. To solve this issue we propose to exclude the data from the first day of training in the regression analyses of the change in performance at the trained task. Our reasoning is that if an item is ambiguous, participants would rapidly learn its expected categorization based on the in-game feedback on accuracy, and then stick to the learned categorization for the remainder of the training. By dismissing the first day of training, we ensure that in the data that will be analyzed, participants would have been already exposed to at least 6 feedbacks on classification accuracy per item and would thus have a stabilized knowledge of each stimulus' category. This approach would have the additional advantage to minimize the confounds related to familiarization with the game

environment, interaction with the tablet, etc. and thus to ensure a better estimate of the evolution of task performance with training. It now reads p. 17: “The first day of intervention will be excluded from the linear regressions of participants' performance to avoid any bias created by the familiarization of the intervention environment (i.e., using the digital screen, familiarization with the training rules, eventual ambiguity in the salty/sweet discrimination).”

My original review mentions one sample t-test to see whether ratings are significantly different to 50 on the VAS. In the response the authors quote a d of 0.2 but this is 0.3 in the manuscript. Please clarify.

We apologize for this typo, the correct effect size was the one reported in the manuscript: 0.3.

In relation to this point I'm not sure I understand the logic here:

To ensure that the healthy and unhealthy items were actually liked by the participants (and not neutral), if a $d \geq 0.2$ difference is observed, participants with the value the farther from the pooled groups median will be successively excluded until the groups are balanced.

Wouldn't a greater d indicate that responses were further from 50 and therefore liked/ disliked and not neutral? If you exclude participants the greater scores you will be excluding the participants who liked/ disliked the foods the most?

Indeed, this was again a typo, we thank the reviewer for having detected it. The illogical statement follows from an inversion of the signs of inequalities in the manuscript. This is now corrected p. 20: “A difference of Cohen's $d > 0.3$ will be expected between the mean palatability ratings of the trimmed trained items and the middle of the analogue scale (i.e., 50 points). To ensure that the healthy and unhealthy items were actually liked by the participants (and not neutral), if a $d \leq 0.3$ difference is observed, participants with the closest value to 50 from the pooled groups will be successively excluded until the groups differ from 50.”

I also don't follow the logic that a similar amount of learning is expected in the control and active groups considering the difference in mapping. If the training does not induce differences in learning then how can this be used as a manipulation check? “Since both the experimental and control training should induce a corresponding amount of learning, I'm again not sure why you would remove participants who show the greatest learning effects – this is potentially very informative and could be an important mediator?”

When referring to “learning” in this section we meant improving basic performance at the trained task (i.e. RT and FA) and not the effect on palatability rating. Yet, when reviewing this section we realized that our initial approach was not optimal. We now propose to assess the difference in learning by comparing between the two groups the slopes of the linear regression between the training days and each participant's performance. It now reads p. 20: “Since both the experimental and control trainings should induce a corresponding amount of learning, a small difference of Cohen's $d < 0.4$ will be expected between the two groups' evolution in performance. These evolution

in performance will be assessed by computing for each participant a linear regression between the training days and the participant's performance (i.e., the score without the impact of the power-ups) and reporting its slope. If a $d \geq 0.4$ difference is observed, participants with the value the farther from the pooled groups median will be successively excluded until the groups are balanced."

Manuscript

Could you please clarify your hypotheses:

H1b): the reverse pattern for the healthy items palatability ratings;

This is slightly ambiguous and could mean a smaller decrease for the exp group or a greater decrease for the control group.

Indeed, we now report the hypotheses in full to p. 5: "H1b): a larger increase in healthy items' palatability ratings in the experimental than the control group between the pre- and post-Intervention."

- H3) the shift in items' valuation mediates the change in weight.

Is this for both unhealthy and healthy shifts i.e. 2 analyses to explore these changes separately? If so please state the 2 hypotheses in full.

As described in the manuscript p. 18, the mediator will be the quotient between the mean unhealthy rating and mean healthy rating to account for a global shift in items' valuation. This has been now clarified in the manuscript p. 6: "H3) the shift in items' valuation (i.e., the quotient between mean unhealthy rating and mean healthy rating) mediates the change in weight."

For the power analysis, can you please provide more detail on the OSF in the form of a readme file. From the current files I couldn't determine whether the data from the 'previous datasets' mentioned in the comment below was available or what the data was (is this your own data or sourced from elsewhere, please provide references if the latter).

2. For each dependent variable separately, we identified the task-related between- and within-subject variance based on previous datasets from studies with the same approaches.

The sources used to determine parameters for our power analyses are now described in the OSF study page "Readme.txt" file in the "SCRIPTS" folder. and in the manuscript p. 7:

- The palatability ratings' between-subject variability were estimated from Lawrence and colleagues (2015b; ref 36);
- The weights' within-subject variability was estimated from the Centers for Disease Control and Prevention (2015; ref 49);
- The weights' between-subject variance is estimated from a paper currently in preparation by our group; it will be made available after the article is published (expected submission within 6 months);

Will the custom-made general health questionnaire be made available, only there is very little information on this?

This questionnaire was on the OSF study page under the name “Pre1) Questionnaire_Santé_Générale” in the “QUESTIONNAIRES” folder. For clarification, we now have updated the name of the questionnaire to “Pre1)GHQ” (i.e. General Health Questionnaire).

Appendix E

UNIVERSITÉ DE FRIBOURG
UNIVERSITÄT FREIBURG

PD Dr Lucas Spierer
University of Fribourg,
Faculty of Science and Medicine,
Medicine Section,
Neurology Unit,
Lucas Spierer & Hugo Najberg
Ch. Du Musée 5
1700 Fribourg, Switzerland

Prof C Chambers
Registered Report Editor
Royal Society Open Science

Dear Prof Chambers, dear Editor,

We hereby would like to submit the stage 2 registered Report by Najberg et al.

We have now included the Result and Discussion section in the manuscript. We could conduct the study as expected and confirm our main hypothesis on the item's devaluation. We did not, however, confirm an effect of the intervention on weight.

We have also highlighted in the Method section the modifications of the Stage 1 part that we had discussed with you before their implementation.

Please do not hesitate to contact us should further information be needed.

Sincerely yours,

Dr Lucas Spierer

Appendix F

RSOS-191288.R3 Response to reviewer

Reviewer 2:

I only have a few comments, as this is a stage 2 registered report. I do not have the time to check their data analyses or perform a detailed comparison of documents. I trust the authors they did this adequately.

I think the most important question raised by the findings how effects can be found on value and not on weight. This raises a number of questions. First, are palatability ratings predictive for consumption? This seems logical but has this been tested before?

The literature on the topic confirms that palatability plays a central role in eating behavior (e.g. PMIDs: 14513063; 8937617; 10336795; 10386914 and for correlational field studies: 10716535; 11006433). We now mention this aspect in the Discussion section, p.30: "This finding may appear surprising given previous evidence for a positive association between food palatability and consumption (e.g., Castro et al., 2000, *Physiol Behav.*, Castro and Dalix, 2000, *Physiol Behav.*). Possible accounts for our finding include the following. [...]"

Second, one explanation for why effects on weight are absent is that participants do not consume the trained products frequently. Can this be ruled out? This could parsimoniously explain why effects are found on value but not on weight. Another explanation could be that participants are not always responsible for the products and amount of food they consume as they may live together with others that are responsible for this. Can this explain the results?

We now mention these alternative accounts in the Discussion section. It reads now p.30: "Third, it might be that the participants did not consume the trained product frequently, even if they rated them as highly palatable. This account is unlikely because one inclusion criteria was to not follow any restrictive diet; the participants thus tended to consume what they liked the most. However, the participants may not have been fully responsible for the food they consume, as e.g. if they live with their parents or friends."

I did not understand the description of the go/no-go task. The healthy foods are not mentioned in this description but are quite crucial. Moreover the difference between the experimental and control conditions is not explained. It is only understandable by looking at the tables.

We have edited the Method section according to the suggestion of the reviewer. It now reads p.13: "In the experimental intervention, healthy items were always Go items and unhealthy NoGo items. In the control intervention, both healthy and unhealthy items had the same chance to be Go or NoGo items. Neutral items had a 20% chance to appear instead of a healthy or unhealthy item in both conditions." and in the description of the Go/NoGo task "The Go category was either sweet items (e.g., orange, ice-cream; half of the training sessions) or salty items (e.g., green beans, hamburger; other half of the sessions), and the NoGo category the other."

Minor points:

Results for weight are not mentioned in the abstract. Moreover, it would be good to make explicit in the abstract how many hypothesis were tested and which were confirmed.

We have edited the abstract according to the comment of the reviewer. It reads now: “We examined whether a one-month intervention combining the practice of Go/NoGo and Cue Approach Training modified the perceived palatability of food items (i.e., decrease of unhealthy and increase of healthy food items’ palatability ratings), and in turn participants’ weights” and “We found a larger decrease of the unhealthy items’ palatability ratings in the experimental (20.6%) than control group (13.1%). However, we did not find any increase of the healthy items’ ratings or weight loss.”

There are a number of paragraphs in the intro that consist of only one sentence.
Page 20 line 8; followed

We have corrected this issue and the typo.